# GATA6 is predicted to regulate DNA methylation in an in vitro model of human hepatocyte differentiation

Takahiro Suzuki [1,2], Erina Furuhata[1,3], Shiori Maeda[1,3], Mami Kishima[1], Yurina Miyajima[1], Yuki Tanaka[1,2], Joanne Lim[1], Hajime Nishimura[1], Yuri Nakanishi[1], Aiko Shojima[1,2] & Harukazu Suzuki [1✉]

Hepatocytes are the dominant cell type in the human liver, with functions in metabolism, detoxification, and producing secreted proteins. Although gene regulation and master transcription factors involved in the hepatocyte differentiation have been extensively investigated, little is known about how the epigenome is regulated, particularly the dynamics of DNA methylation and the critical upstream factors. Here, by examining changes in the transcriptome and the methylome using an in vitro hepatocyte differentiation model, we show putative DNA methylation-regulating transcription factors, which are likely involved in DNA demethylation and maintenance of hypo-methylation in a differentiation stage-specific manner. Of these factors, we further reveal that GATA6 induces DNA demethylation together with chromatin activation in a binding-site-specific manner during endoderm differentiation. These results provide an insight into the spatiotemporal regulatory mechanisms exerted on the DNA methylation landscape by transcription factors and uncover an epigenetic role for transcription factors in early liver development.

[1] Laboratory for Cellular Function Conversion Technology, RIKEN Center for Integrated Medical Science (IMS), RIKEN Yokohama Campus, 1-7-22 Suehiro-cho, Tsurumi-ku, Yokohama City, Kanagawa 230-0045, Japan. [2] Graduate School of Medical Life Science, Yokohama City University, 1-7-29 Suehiro-cho, Tsurumi-ku, Yokohama City, Kanagawa 230-0045, Japan. [3] These authors contributed equally: Erina Furuhata, Shiori Maeda. ✉email: harukazu.suzuki@riken.jp

Hepatocytes, the major parenchymal cells in the liver, are responsible for key liver functions such as metabolism and detoxification. In embryogenesis, the first fate decision to the hepatocyte lineage is the differentiation of primitive streak cells to definitive endoderm (DE) cells, which are a common precursor of endoderm tissues such as the liver, pancreas, and gut. Hepatoblasts are hepatic progenitor cells derived from the DE cells, which then sequentially differentiate into fetal-like hepatocytes and mature hepatocytes. Thus, hepatocytes emerge from pluripotent stem cells through several progenitor cell types.

Several transcription factors (TFs), including c-Jun, members of the Hepatocyte Nuclear Factor (HNF), and GATA family genes, are known to play important roles in liver development and hepatocyte differentiation[1–5]. Notably, GATA6 knock-out mice die around E5.5 due to a deficiency of extra-embryonic endoderm development, which can be rescued by tetraploid embryo complementation assays, indicating that GATA6 is required for liver development and hepatic specification[3,5]. Thus, multiple TFs sequentially and coordinately regulate peripheral genes necessary for hepatocyte differentiation.

Gene expression dynamics are regulated not only by the action of transcription factors but also by epigenetic modifications such as DNA methylation. DNA methylation of gene regulatory regions appears to be associated with silencing the expression of the downstream gene[6]. Therefore, the DNA methylation profile is dramatically altered during embryogenesis and cellular differentiation, with roles in tightly regulating the expression of downstream genes[7]. Indeed, DNA methylation plays a crucial role in the expression of numerous liver-specific genes[8,9], and DNA methyltransferase (DNMT) inhibitors facilitate trans-differentiation of adipose tissue-derived stem cells or mesenchymal stem cells to hepatocyte-like cells (HLCs)[10,11]. Collectively, these findings show that DNA methylation is a crucial factor for hepatic differentiation.

The gain of DNA methylation is directly achieved by de novo DNMTs and is maintained during cell divisions by a maintenance DNMT. On the other hand, DNA demethylation is achieved by cell proliferation dependent passive DNA demethylation[12] or active DNA demethylation based on sequential oxidative processes by ten-eleven translocation (TET) enzymes[13], followed by base-excision repair[14]. In addition, the oxidized forms of methylated cytosine (5-hydroxymethyl cytosines (5hmC), 5-formyl cytosine (5fC), and 5-carboxy cytosine (5caC)) are also depleted by passive demethylation mechanisms as these bases are not recognized by the maintenance DNA methylation mechanism[15]. Thus, DNA methylation is a balance between the gain and loss of methylated bases.

In addition to the mechanisms by which DNA methylation is gained and lost, mechanisms underlying spatiotemporal regulation of DNA methylation are also critical in understanding the overall dynamics of DNA methylation. We and other groups recently reported that some TFs regulate the timing and site-specificity of DNA demethylation[16–18]. Thus, a growing body of evidence suggests critical roles for TFs in the regulation of DNA methylation. However, the epigenetic roles of TFs specific for hepatocyte differentiation are yet to be identified.

In the present study, we combine TF binding motif (TFBM) over-representation analysis for differentially methylated regions with transcriptome analysis. We identified TFs with putative roles in regulating DNA methylation during hepatocyte differentiation by studying in vitro process of hepatocyte differentiation from human induced pluripotent stem cells(iPSCs). Of these TFs, we demonstrated that GATA6 is a master regulator for DNA demethylation and chromatin activation during the differentiation of the DE. Our data provide important insights into the regulatory mechanisms shaping the DNA methylation landscape during hepatocyte differentiation.

## Results

**Evaluation of an in vitro hepatocyte differentiation model.** We induced HLCs from human iPSCs in vitro using the Cellartis hepatocyte differentiation system (Takara bio), which is composed of three differentiation steps: iPSC to DE-like cell, DE-like cell to hepatoblast-like cell, and hepatoblast-like cell to HLC, followed by a maintenance culture (Fig. 1a, b). It has been reported that although this hepatic differentiation protocol does not activate several hepatic function-related genes, the efficiency of the hepatic specification is relatively high[19,20]. Indeed, over 96% of the cells in days 21 and 28 cultures were HNF4α positive without pancreatic or cardiac cell marker expressing cells (Fig. 1c, d and Supplementary Fig. 1a). Tissue-specific gene enrichment analysis for the upregulated genes between day 0 and day 28 revealed enrichments of endodermal tissue, such as the liver, supporting the hepatic differentiation (Supplementary Fig. 1b). To evaluate the differentiation in detail, we analyzed the mRNA expression of differentiation markers. Concurrent with the decrease of the pluripotent markers, DE markers peaked at day 7, indicating the DE-like cell stage (Supplementary Fig. 1c). On day 14, hepatoblast markers were upregulated, indicating the hepatoblast-like cell stage (Supplementary Fig. 1c). Between day 14 and day 21, hepatic markers were upregulated, and a part of them kept the high expression between day 21 and day 28, although others were decreased (Supplementary Fig. 1c). *Alpha-fetoprotein (AFP)*, produced by fetal livers but not by adult livers, increased to 4.6-fold greater expression than the published fetal liver CAGE expression data[21] between day 7 and day 21. Then *AFP* expression decreased to 49.2% of the fetal liver level from day 21 to day 28 (Supplementary Figs. 1c, 2a). AFP protein was also detected by immunocytochemistry of day 21 cells and Enzyme-linked immunosorbent assay in the culture medium (Fig. 1e and Supplementary Fig. 2b). Whereas, although expression of *Albumin (ALB)*, which is expressed in mature hepatocytes, was increased upon differentiation from day 14, it was much lower than that of the published hepatocyte, fetal liver, and adult liver (Supplementary Figs. 1c, 2a). Thus, these results suggest that day 21 and 28 cultures correspond to the fetal or immature hepatocyte stage, consistent with the earlier report[19]. Collectively, although the day 28 cells have the drawback of several mature hepatic gene expressions, the in vitro hepatocyte differentiation recapitulates the in vivo liver development until the stage of fetal or immature hepatocytes.

**DNA methylation dynamics throughout hepatocyte differentiation.** To investigate changes in DNA methylation during the in vitro hepatocyte differentiation model, we performed a methylome analysis of the time-course samples. Hierarchical clustering showed that iPSCs and DE-like cells were segregated from the differentiated cells that followed in the time-course, consistent with a commitment to the hepatocyte lineage (Fig. 1f). Comparing adjacent time points, we identified 3088, 446, 38, and 54 methylated CpGs and 3809, 11652, 7383, and 864 demethylated CpGs in each interval, indicating the bias toward demethylation (loss of methylation) (Fig. 1g). The expression of DNA methyltransferases tended to decrease with differentiation, in line with the decline in the number of methylated probes (Supplementary Fig. 2c). While the expression of *TET1* decreased with differentiation, that of *TET2* spiked on day 14, when the number of identified demethylated probes was the highest. Thus, these data suggest that the bias toward demethylation depends on the balance of methylation and demethylation enzymatic activities.

We associated biological functions to the differentially methylated regions using the Genomic Region Enrichment of Annotations Tool (GREAT)[22] and summarized the results based

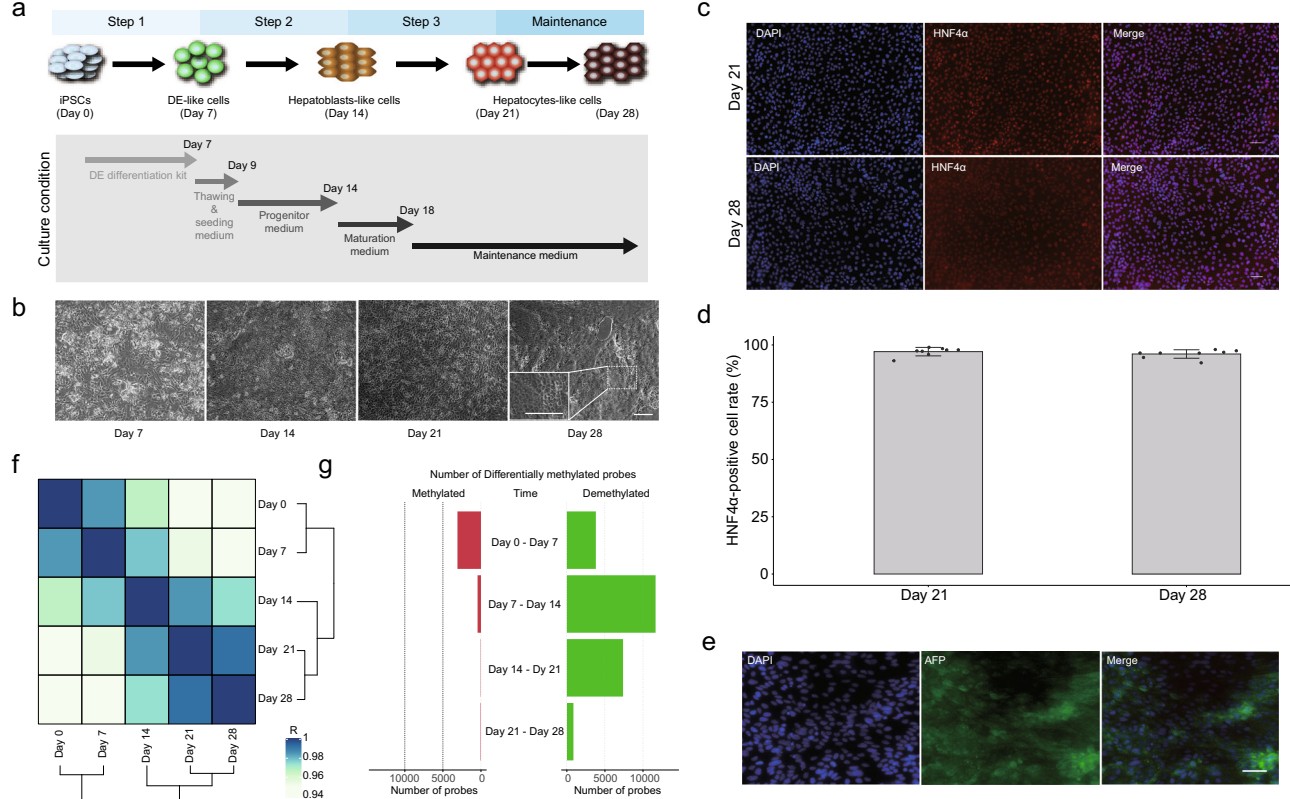

**Fig. 1 Time-course methylome analysis of hepatocyte differentiation. a** Schematic illustration of in vitro hepatocyte differentiation and culture condition of each step. **b** Representative micrographs of cells at each time point. The scale bar is 100 μm. At the bottom left of day 28, a higher magnification image of the area surrounded by the doted-square shows. **c** Immunocytochemistry for HNF4α. The scale bar is 50 μm. **d** Bar plot showing the average rate of HNF4α stained nuclei. The error bar is s.d. Dots represent the individual data. The data were collected from four different fields of a culture well in two biological replicates. **e** Immunocytochemistry for AFP. The scale bar is 50 μm. **f** A correlation matrix with hierarchical clustering. The color represents the correlation coefficient (R). **g** The number of differentially methylated probes.

on semantic similarity[23]. This analysis revealed an enrichment in development and morphogenesis-related Gene Ontologies (GOs), including "pattern specification process", "anatomical structure development", "radial pattern formation", "developmental process", and "regulation of developmental process" (Supplementary Fig. 3). Overall, these results imply that DNA methylation mainly regulates genes related to the developmental process, consistent with specifying the cells into the hepatocyte lineage.

**Prediction of DNA methylation-regulating transcription factors throughout hepatocyte differentiation**. To identify TFs that regulate binding site-directed DNA methylation (hereafter referred to as DNA methylation-regulating TFs), we performed TFBM over-representation analysis for the differentially methylated CpG regions between two adjacent time points of the differentiation time-course. Because some TFs, such as TFs in the same family, share the same or similar binding motif, the results of TFBM over-representation analysis often include false positives. Therefore, to reduce the possibility of false positives, we further narrowed down the overrepresented TFBMs by considering TF expression (CAGE tag-per-million (TPM) ≥50) in either of the two adjacent time points of an interval (Fig. 2a). Comparing each adjacent time point, we identified in total 16 putative DNA methylation-regulating TFs in the methylated regions. Of these, 13 TFs, including POU5F1, a pluripotent cell-specific TF, were identified between Day 0 and Day 7 (Fig. 2b). In addition, GATA6, GATA3, and GATA4 were identified between day 7 and day 14 (Fig. 2b). Interestingly, these putative DNA methylation-regulating TFs for the methylated regions were

prone to being highly expressed in the earlier time point of the intervals and then declined with the progress of differentiation (Fig. 2b, c and Supplementary Fig. 4a).

On the other hand, we identified 50 putative DNA methylation-regulating TFs in demethylated regions. Of these, *HNF4A*, an essential TF for liver development, was identified between day 7 and day 14, and between day 14 and day 21 (Fig. 2d). In addition, the over-representation of TFBMs for activator protein 1 (AP-1) components, such as JUN and FOS, which are involved in stress response and regeneration in the liver[24], increased from day 14 to day 21 (Fig. 2d). Importantly, GATA6, GATA4, and GATA3, which were overrepresented in the regions methylated between day 14 to day 21, were first overrepresented in the DE-like cell differentiation stage, and the over-representation of these binding motifs declined as differentiation proceeded (Fig. 2d). Contrary to the putative DNA methylation-regulating TFs for the methylated regions, expression of the putative DNA methylation-regulating TFs for the demethylated regions tends to be upregulated in later time points of the intervals (Fig. 2e and Supplementary Fig. 4b). Taken together, these results suggest that diverse TFs cooperatively regulate the DNA methylation landscape. In particular, GATA transcription factors appear to be the major factors for the regulation of DNA methylation, participating in both methylation and demethylation changes.

**GATA6 regulates binding site-directed DNA demethylation.** Of the GATA proteins, GATA4 and GATA6 are known to be essential TFs for the DE differentiation[2,3]. Therefore, we focused

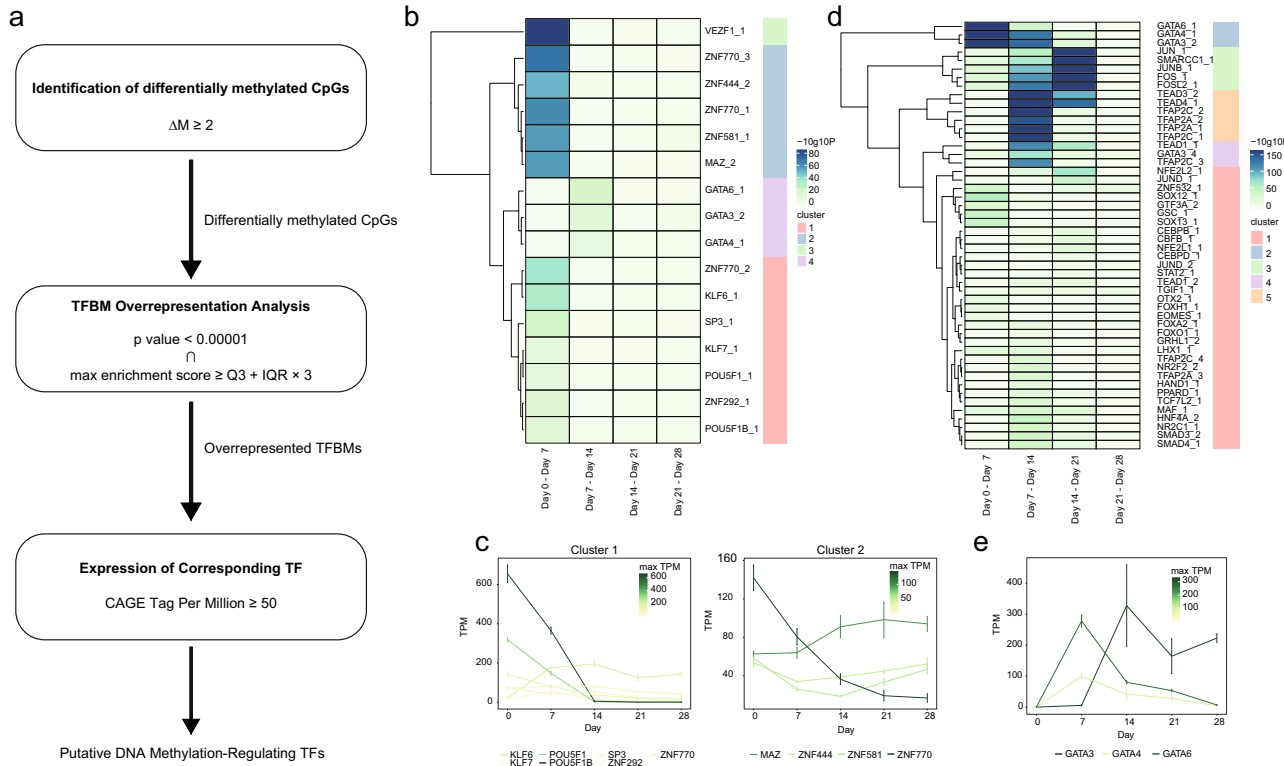

**Fig. 2 Prediction of DNA methylation-regulating TFs. a** The workflow of DNA methylation-regulating TF prediction. **b, d** Heatmap showing the *P* value of the one-sided (greater) exact Poisson test of overrepresented TF binding motifs at methylated (**b**) and demethylated (**d**) regions. Each column is an interval of adjacent time points. Each row is a putative methylation-regulating TF. The dendrogram of hierarchical clustering is shown at the left of the heatmap, and clusters are shown at the right of the heatmap as colors. **c** mRNA expression profile of the cluster 1 and 2 putative DNA methylation-regulating TFs for methylated regions. X- and Y-axes show time points of differentiation (hours from differentiation initiation) and tag-per-million (TPM) of CAGE, respectively. The color of each line and the error bar represent the maximum TPM and s.d, respectively. The experiment was performed in three biological replicates. mRNA expression profiles of the clusters 3 and 4 putative DNA methylation-regulating TFs for methylated regions are shown in Supplementary Fig. 4. The experiment was performed in three biological replicates. **e** mRNA expression profile of the GATA3, GATA4, and GATA6. X- and Y-axes show time points of differentiation (hours from differentiation initiation) and tag-per-million (TPM) of CAGE, respectively. The color of each line and the error bar represent the maximum TPM and s.d, respectively. The experiment was performed in three biological replicates.

the following analysis on possible epigenetic functions of GATA4 and GATA6 in DE differentiation. First, we performed qRT-PCR to confirm the expression changes of *GATA4* and *GATA6* during the DE-like cell differentiation. *GATA6* and *GATA4* expression increased at 48 and 54 h, respectively, after the induction of differentiation and continued to increase with differentiation (Fig. 3a). Furthermore, *GATA6* expression increased drastically, greater than 1000-fold at 60 h compared with 48 h, whereas *GATA4* expression increased only fourfold at 66 h compared with 48 h, indicating the dominant impact of GATA6 (Fig. 3a). Indeed, GATA6 is reported to be an upstream factor of GATA4[25].

Because our recent screening study identified GATA6 as a candidate for DNA demethylation-regulating TF[26], we performed Cloning-based bisulfite sequencing to validate the screening result. Cloning-based bisulfite sequencing for ±100 bp regions from four CpGs which were demethylated by GATA6 overexpression in the earlier report[26] showed that turning the GATA6 overexpression off tended to partially recover the GATA6 overexpression-induced DNA demethylation (Fig. 3b and Supplementary Fig. 5a). A pull-down assay between HaloTag-fused TET proteins and GATA6 revealed an association between TET proteins and GATA6, suggesting that GATA6 recruits TET proteins to their binding sites (Fig. 3c and Supplementary Fig. 5b). Thus, these results suggest that GATA6 induces DNA demethylation, recruiting the TET proteins.

**DNA demethylation accompanies GATA6 binding during iPS-DE-like cell differentiation.** To investigate the dynamics by which GATA6 regulates DNA demethylation, we performed finer time-course transcriptome and methylome analyses during the time window of GATA6 emergence (after 0, 48, 54, 60, 66, and 72 h of the differentiation process) (Fig. 4a). T, a marker of the primitive streak, was upregulated at 48 h and downregulated after 54 h (Supplementary Fig. 6a). DE markers were upregulated during the period of 48 to 72 h (Supplementary Fig. 6a). In agreement with the qRT-PCR analysis (Fig. 3a), the expression of GATA6 was slightly upregulated at 48 h and drastically increased after 48 h (Supplementary Fig. 6a). Hence, our data indicate that DE commitment occurs during the period of 48 to 72 h.

By comparing adjacent time points, we identified 120 (0 to 48 h), 94 (48 to 54 h), 26 (54 to 60 h), 19 (60 to 66 h), and 50 (66 to 72 h) methylated CpGs and 220 (0 to 48 h), 226 (48 to 54 h), 33 (54 to 60 h), 27 (60 to 66 h), and 27 (66 to 72 h) demethylated CpGs, respectively (Fig. 4b). However, we did not find the GATA6 binding motif overrepresented in those demethylated regions during any interval (Supplementary Fig. 6B). Because the time intervals between adjacent time points are 6 h except for the initial period (0 to 48 h), the changes in methylation levels may not be enough to be detected as demethylation (ΔM > 2). Indeed, the GATA6 binding motif was overrepresented at the regions demethylated between 0 and 72 h, and these demethylated regions

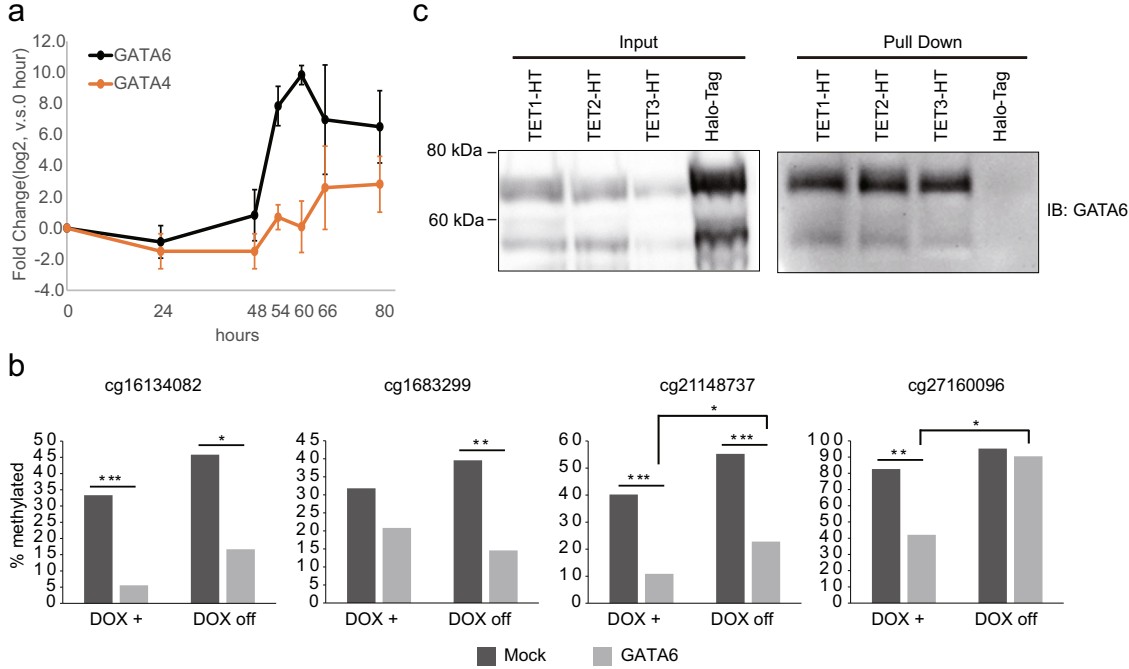

**Fig. 3 GATA6-mediated binding site-directed DNA demethylation. a** qRT-PCR analysis for GATA4 (orange) and GATA6 (black). X- and Y-axes show time points of differentiation (hours from differentiation initiation) and fold-change (compared with 0 h, $\log_2$ scaled), respectively. The error bar is s.d. The experiment was performed in three biological replicates. **b** Percentage of DNA methylation of ±100 bp regions from four selected demethylated CpGs, analyzed by cloning-based bisulfite sequencing. Mock is mock control transduced and GATA6 is the doxycycline-inducible GATA6-transduced HEK293T cells. The cells were treated with doxycycline for two weeks (DOX+) and then cultured for 2 more weeks without doxycycline (DOX off). (*$p < 0.05$; **$p < 0.01$, ***$p < 0.001$; One-sided Fisher's exact test). **c** HaloTag pull-down assay in HaloTag-fused TET1 (TET1-HT) and GATA6 co-overexpressing HEK293T cells, in HaloTag-fused TET2 (TET2-HT) and GATA6 co-overexpressing 293T cells, in HaloTag-fused TET3 (TET3-HT) and GATA6 co-overexpressing HEK293T cells, and in Control HaloTag (HaloTag) and GATA6 co-overexpressing HEK293T cells followed by immunoblotting with GATA6 antibody.

tend to be continuously demethylated from 0 h (Supplementary Fig. 6c, d). Therefore, to investigate whether the GATA6 binding motif is overrepresented for the cumulative changes in methylation, we compared the regions demethylated at each time point with that at 0 h. We identified 220 (0 to 48 h), 236 (0 to 54 h), 416 (0 to 60 h), 876 (0 to 66 h), and 620 (0 to 72 h) demethylated CpGs (Fig. 4c). Because these demethylated CpGs include those that were demethylated in the earlier time point and maintained the hypomethylated status, we only selected the demethylated CpGs that were newly detected as demethylated CpGs at each time point (referred to as uninherited demethylated CpGs) to clarify the effects of each additional period. GATA6 motif over-representation analysis in the vicinity of these uninherited demethylated CpG (uninherited demethylated regions: UDRs) revealed the GATA6 binding motif was overrepresented at 0 to 60 h and 0 to 66 h (Fig. 4d). To further substantiate the over-representation of the GATA6 binding motif at the UDRs, we performed ChIPmentation, which can provide evidence for actual physical interactions between genomic regions and GATA6[27]. Consistent with the expression pattern of GATA6, GATA6 binding was not enriched at UDRs during the period 0 to 48 h, indicating the irrelevance of GATA6 during this period (Fig. 4e). In contrast, unlike binding motif over-representation, ChIPmentation showed interactions between GATA6 protein and most of the UDRs of all comparisons apart from the 0 to 48 h, consistent with the expression pattern of the GATA6 (Fig. 4e and Supplementary Fig. 6a). The GATA6 ChIPmentation peaks at the 72 h significantly overlapped with the regions demethylated during DE-like cell differentiation (486 regions: $P$ value = 0.001, one-sided permutation test). Furthermore, of the overlapped regions, 48.8% (237 regions: $P$ value = 0.001, one-sided permutation test) overlapped with the demethylated region by

GATA6 overexpression[26] (Supplementary Fig. 6e and Supplementary Data 1). Thus, our results indicate a correlation between GATA6 binding and DNA demethylation during DE-like cell differentiation.

**The interrelation between DNA demethylation and chromatin status during iPS-DE differentiation.** The majority of the demethylated regions were not promoters but other types of regulatory regions such as enhancers and non-annotated regulatory regions (Supplementary Fig. 7a). Therefore, we investigated the chromatin status of the demethylated regions. Active regulatory regions transcribe several classes of transcripts, including mRNA, promoter-upstream transcripts (PROMPTs), and enhancer RNAs (eRNAs), which are typically transcribed within ± 250 bp from the center of the regulatory region[28]. Thus, the transcription level serves as an indicator of chromatin activity. To investigate the chromatin activity of the demethylated regions, we measured the average TPM of the UDRs (± 250 bp regions from the uninherited demethylated CpGs) by CAGE. The average TPMs of the UDRs were prone to increase as differentiation proceeds in all comparisons except for the 0 to 48 h, indicating the activation of gene regulatory regions (Fig. 5a).

To further analyze the interrelation between GATA6-mediated DNA demethylation and chromatin status, we measured chromatin accessibility by Omni-ATAC-seq[29]. Chromatin accessibility at the UDRs increased between 0 and 48 h and was maintained over the following time points at most of the demethylated regions (Fig. 5b), in agreement with the transcription pattern and GATA6 binding (Fig. 4e and Supplementary Fig. 6a). These demethylated regions coincident with chromatin

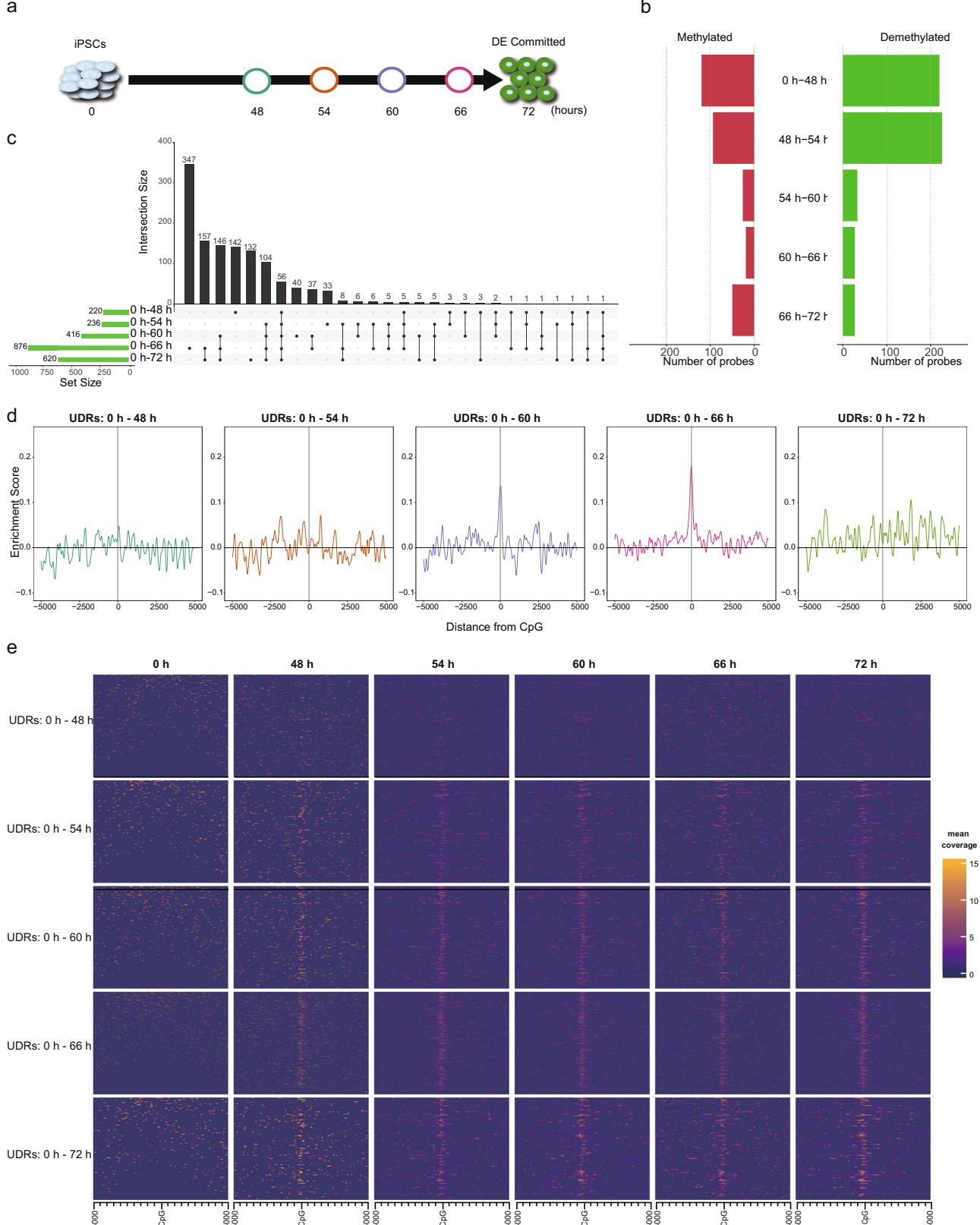

opening during the DE differentiation stage included regulatory regions of known GATA6 targets, such as SOX17 and GATA6 (autoregulation) (Fig. 5c and Supplementary Fig. 7b)[30]. Notably, the demethylated regions noted during DE-like cell differentiation were only marginally accessible in iPS cells (0 h), although

GATA6 is not expressed at that time, suggesting that target regions of the GATA6-mediated DNA demethylation are predefined by chromatin accessibility (Fig. 5b). We also investigated the change in chromatin accessibility of ATAC-seq peaks, which opened in the period 0 to 48 h, dividing the peaks into

**Fig. 4 GATA6-mediated DNA demethylation analysis during DE differentiation. a** Schematic illustration of time-course sampling of DE differentiation. **b** The number of differentially methylated probes. **c** UpSet plot showing the demethylated probes at each comparison. The vertical bars indicate the number of intersecting demethylated probes between comparisons, denoted by the connected black circles below the histogram. The horizontal bars show the demethylated probe set size. **d** Distribution of enrichment score for the GATA6 binding motif within ±5000 bp of demethylated CpG probes at each time point compared with undifferentiated iPS cells (0 h). X- and Y-axes show distance from probe CpG position and enrichment score, respectively. Horizontal and vertical lines are enrichment score = 0 and demethylated CpG position, respectively. The colors of each plot represent the colors of time points shown in (**a**). **e** Enrichment heatmap showing mean GATA6 ChIPmentation read coverage of 100 bp window at a range of ±5 kbp from demethylated CpGs. Each time point is horizontally aligned, and each of the UDRs is vertically aligned. Dark blue is low coverage and orange is high coverage.

demethylated and hyper-methylation-maintaining regions. Mean accessibility level was higher in peaks at the demethylated regions than those at the hyper-methylation-maintaining regions at all time points even before chromatin opening (iPSCs), and the deviation of chromatin accessibility tended to be higher in the hyper-methylation-maintaining regions, suggesting that hypo-methylation stabilizes the open chromatin status (Supplementary Fig. 7c).

Taking advantage of our time-course multi-omics dataset, we compared the kinetics of GATA6 expression, GATA6 binding to the genome (ChIPmentation), methylation change (M-value), and chromatin status (ATAC-seq and Transcript) (Fig. 6a). Overall, the kinetics of GATA6 binding, chromatin accessibility, and transcription observed the same trends, regardless of the UDRs. While transcription levels and −ΔM value at the UDRs tended to increase after 48 h in accordance with GATA6 expression, GATA6 binding dramatically increased between 48 to 54 h and plateaued at 54 h, with a transient decrease at 66 h. Of note, chromatin accessibility increased in the period 0 to 48 h or 54 h and then decreased after the peaking, although DNA continued to be demethylated.

## Discussion

In the present study, by applying transcriptome and TFBM over-representation analyses for differentially methylated regions, we comprehensively identified putative DNA methylation-regulating TFs during hepatocyte differentiation. Of these TFs, our results provide multiple strands of evidence that GATA6 is a primary epigenome regulator for the iPSC to DE-like cell differentiation.

Expressions of several hepatic function-related genes, such as Albumin, were low or not even on the day 28 HLCs (Supplementary Figs. 1c, 2a). Furthermore, several hepatic markers were decreased between day 21 and day 28, when the cells were cultured in the maintenance medium. This expression decline is likely the same phenomenon as the rapid loss of hepatic functionality when primary hepatocytes is cultured ex vivo. These drawbacks were already pointed out in the previous reports[19,20]. Thus, because the liability must be due to the differentiation protocol, the functionality may be enhanced by recently reported improved protocol or 3D organoid culture[31,32]. Nevertheless, because hepatic differentiation was high efficiency and near-homogeneous (Fig. 1c, d), the protocol used in this study seems to recapitulate the key molecular events of hepatic differentiation, which fulfill the purpose of this study.

We found enrichment of many TFBMs at demethylated regions during hepatocyte differentiation correlated with the expression of corresponding TFs (Fig. 2d, e and Supplementary Fig. 4b). In contrast, some TFBMs, such as POU5F1, GATA4, and GATA6, were overrepresented mainly in the methylated regions, and the expression of the corresponding TFs was inversely correlated with methylation change (Fig. 2b, c, e and Supplementary Fig. 4a), suggesting that gain of methylation may result from the loss of hypo-methylation maintenance by DNA demethylating-TFs. Interestingly, GATA4 and GATA6 binding motifs are also

overrepresented in the demethylated regions at the DE-like cell differentiation, showing the dual roles of GATA4 and GATA6. To summarize, our data suggest that TF-mediated regulation of DNA methylation acts in both the gain and loss of methylation.

HNF4A is required during liver development for the establishment of 5hmC via interactions with TET3[33]. Although the methylation array analyses used in the present study do not distinguish between methylated cytosine and 5hmC, HNF4A binding motifs were overrepresented in the demethylated regions during the hepatoblast-like cell differentiation (Fig. 2d). Since 5hmC has a short half-life[34], our results suggest that HNF4A-induced 5hmC is immediately converted to 5fC, 5caC, or unmodified cytosine. Thus, these issues are unresolved at present and require further investigations. Concomitantly with the mRNA expression, GATA6 proteins bound to the vast majority of demethylated regions, which was maintained through differentiation (Fig. 4e and Supplementary Fig. 6a). Ectopic expression of GATA6 in HEK 293 T cells supported the GATA6-mediated binding site-directed DNA demethylation (Fig. 3b)[26]. As GATA6 protein is associated with TET proteins, recruitment of TET protein to the GATA6 binding sites may be one of the mechanisms underlying the demethylation of RUNX1[18]. Thus, these results demonstrate that GATA6 is a crucial regulator of DNA demethylation for early hepatic development.

GATA6 motif over-representation in the demethylated regions was not completely consistent with ChIPmentation results. Since GATA-binding proteins can bind various non-canonical motifs with comparable affinities to the canonical GATA-binding motif[35], our TFBM over-representation analysis, in which we used the canonical GATA6 motif, may underestimate the TF binding. In addition, ChIPmentation may include indirect binding of GATA6 via their co-factors such as Friend Of GATA (FOG) proteins[36]. Nevertheless, TFBM over-representation has a value in predicting TF binding, because it does not depend on experimental difficulties such as antibody quality.

GATA6 is reported to be a pioneer factor that directly binds to non-permissive heterochromatin and primes the opening of chromatin and histone modifications by interacting with the chromatin remodeling complex[37]. DNA demethylation by GATA6 may be a step toward pioneering. On the other hand, how the pioneer factors recognize their target regions is not clear yet[38]. Our finding that GATA6 binding regions were already slightly accessible in iPSCs may explain the mechanism (Fig. 5b).

Chromatin accessibilities at the DNA demethylated regions increased from 0 to 48 h or 54 h and then declined, although DNA methylation kept decreasing, inconsistent with the notion that DNA methylation is correlated with closed chromatin. As the chromatin accessibility assay reflects not only the presence of open chromatin or nucleosome density but also TF binding, this data may be due to the TFs binding.

Although the underlying molecular mechanisms have not been investigated in this study, our analysis proposes a sequential reaction coordinated with the expression pattern of TFs. DNA demethylating-TFs first bind to the permissive heterochromatin sites where the TFBM are located. They then open

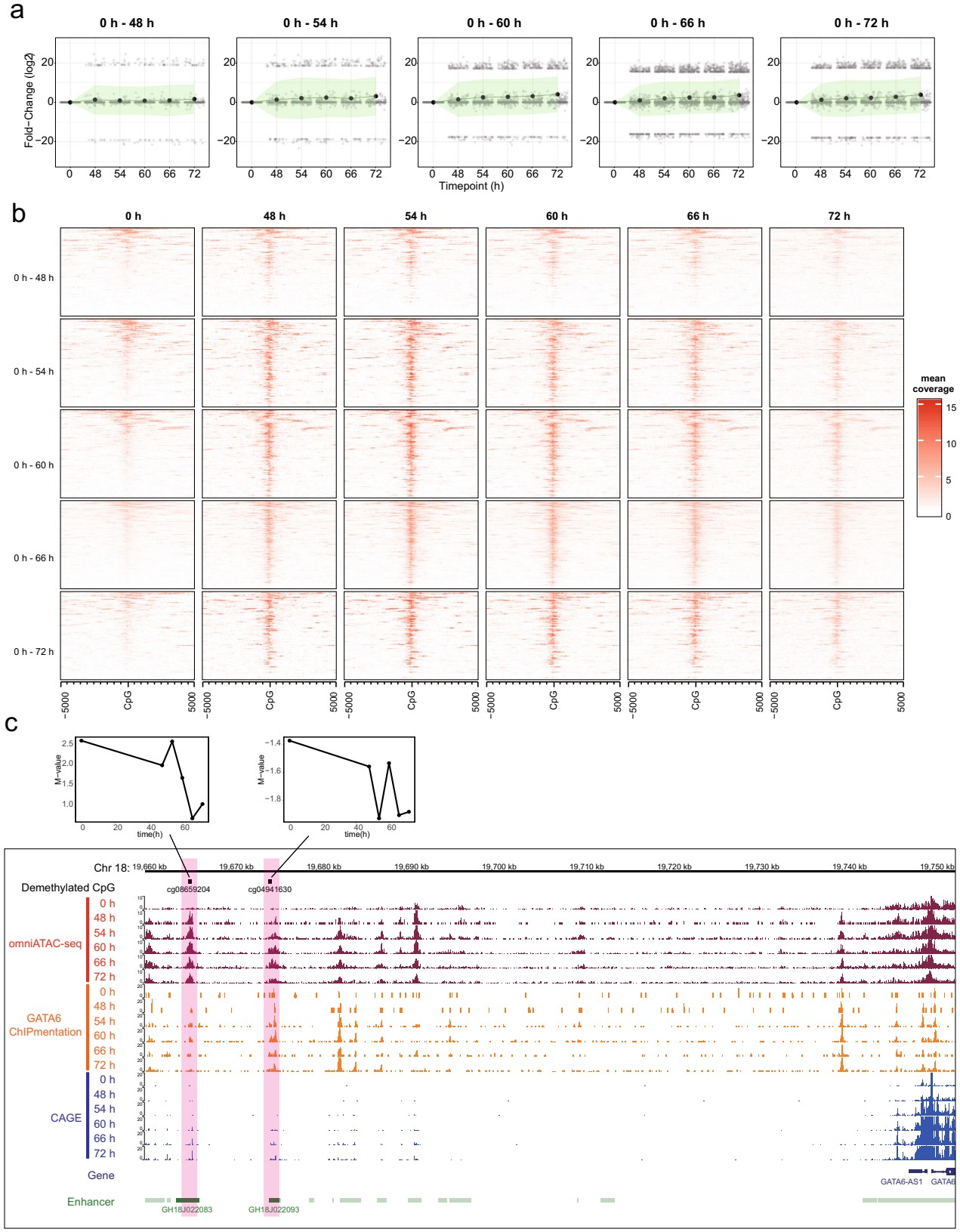

and activate the chromatin at the binding sites, and finally complete DNA demethylation (Fig. 6b). This sequential reaction may be merely due to differences in reaction times between chromatin remodeling and DNA demethylation. While chromatin remodeling is an enzymatic reaction, DNA demethylation

is achieved by several mechanisms, including cell division-dependent passive DNA demethylation that takes more time than a single enzymatic reaction.

GATA6 plays pivotal roles in endoderm cell development and pancreas and lung formations[39–42]. Therefore, GATA6

**Fig. 5 Chromatin status at demethylated regions. a** Change in average TPM of demethylated regions during DE differentiation. X- and Y-axis represents time point and relative TPM (vs. TPM of 0 h), respectively. The black and gray circles represent average and individual data points, respectively. The light-green shade is the standard deviation. **b** Heatmaps showing mean Omni-ATAC-seq read coverage of 100 bp window at a range of ±5 kbp from demethylated CpGs. Each time point is horizontally aligned, and each of the UDRs is vertically aligned. Red is higher coverage of Omni-ATAC-seq reads. **c** A screenshot of the genome browser showing DNA demethylated regions during the DE differentiation stage, Omni-ATAC-read coverage, GATA6 ChIPmentation read coverage, and CAGE read coverage at GATA6 upstream region. The scale of each dataset is coverage of 10 million 100 nt reads. Red translucent rectangles represent demethylated regions. The enhancer track is based on the GeneHancer database, and enhancers overlapped with the demethylated region are shown as dark green. The genome version is hg19. M-value profiles of each demethylated probe are shown above (x-axis: time point (hour), y-axis: M-value).

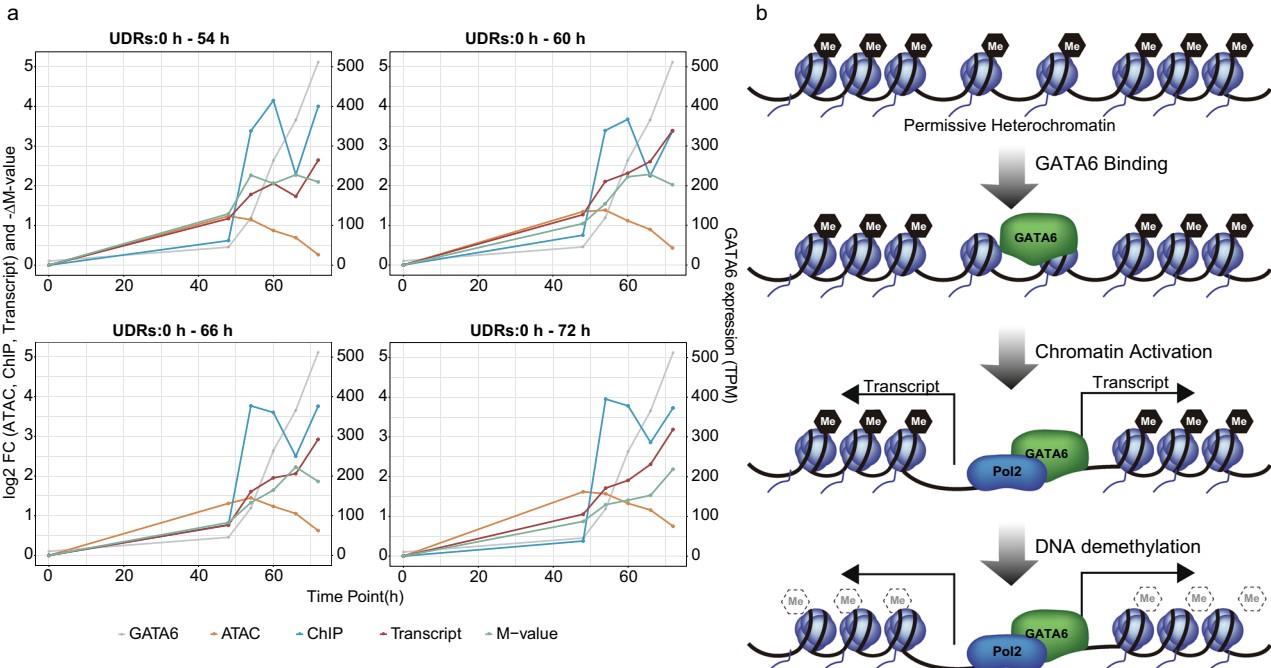

**Fig. 6 Multi-omics kinetic analysis. a** Line plots showing changes in each demethylated region's omics data. Y-axis is $\log_2$ fold-change (FC) for read coverages of ATAC-seq and ChIPmentation for GATA6 (left scale), and -ΔM-value (left scale); TPM for GATA6 expression (right scale). X-axis represents the time points of the differentiation. CAGE-based GATA6 expression profiles are identical among the panels. **b** A schematic illustration showing a model of interrelation between GATA6-mediated DNA demethylation and chromatin status.

haploinsufficiency causes several diseases, such as pancreatic agenesis[43]. Because the epigenetic function of GATA6 was also confirmed in HEK293T cells, which are non-endodermal, by the artificial ectopic expression system, GATA6-mediated DNA demethylation may be associated with the other biological system and pathology of these diseases.

## Methods

**Cell culture and in vitro differentiation**. The 201B7 human iPS cell line[44] was acquired from the RIKEN BioResource Center (BRC) and was cultured in a Cellartis® DEF-CS™ Culture System (Takara Bio Inc., Shiga, Japan). For in vitro hepatocyte differentiation and DE differentiation, we used the Cellartis® Hepatocyte Differentiation Kit (Takara Bio Inc.) and the Cellartis® DE Differentiation Kit (Takara Bio Inc.), respectively, according to the manufacturers' instructions. The culture conditions are shown in Fig. 1a.

**Immunocytochemistry**. The cells cultured on a cover glass were fixed in 4% formaldehyde for 15 min, followed by blocking using 5% skim milk. The cells were then incubated with primary antibodies diluted by the antibody reaction buffer (1% BSA 0.2 % Triton-X100 containing D-PBS(+/+)) for 12 h at 4 °C. After washing in D-PBS(+/+) twice, the cells were incubated with secondary antibodies diluted by the antibody reaction buffer for 1 h at RT. The cells were mounted in slow-fade (Thermo Fisher Scientific Inc., Waltham, MA, USA) and analyzed by a BZ-X810 fluorescent microscope (Keyence Corporation, Osaka, Japan). Cell number measurements based on the immunocytochemistry images were performed using ImageJ. The antibodies used for the immunocytochemistry are shown in Supplementary Table 2.

**Methylation array analysis**. Genomic DNA was isolated using a NucleoSpin® Tissue Kit (Macherey-Nagel, Düren, Germany). The methylation array used an Infinium Human methylationEPIC BeadChip (Illumina, San Diego, CA, USA), according to the manufacturer's instructions. Data normalization and M-value computation were performed *lumiMethyNorm* implemented in the InfiniumDiffMetMotR R package. Differentially methylated probes were identified as those with an M-value difference (ΔM) > 2.

**Cap analysis gene expression**. Total RNA was extracted using NucleoSpin® RNA (Macherey-Nagel) and 3 µg of the total RNA were reverse-transcribed using superscript III (Thermo Fisher Scientific Inc.). Cap structure of the RNA was biotinylated, followed by treatment of RNase ONE ribonuclease (Promega Corporation, Madison, WI, USA), RNA-cDNA hybrids were captured using streptavidin-coated magnetic beads (Thermo Fisher Scientific Inc.), and only single-stranded cDNAs were released from the beads. The released cDNAs were ligated to linkers and the second strand were synthesized using Deep Vent (exo-) DNA polymerase (New England BioLabs, Ipswich, MA, USA). The CAGE libraries were sequenced using single-end reads of 50 bp on the Illumina HiSeq 2500 (Illumina). The extracted CAGE tags were then mapped to the human hg19 genome by STAR. The tags per million (TPM) were calculated for each FANTOM5 TSS peak and regions extended ±250 bp from each differentially methylated CpG. Gene expression levels of each gene were computed as the sum of multiple TSS peaks associated with a single gene. The CAGE analysis was performed in three biological replicates.

**Omni-ATAC-seq**. Ten southland cells were stored at −80 °C in STEM CELL-BANKER® (Takara Bio Inc.) until use. The cells were washed with PBS and nuclei were extracted. The extracted nuclei were resuspended in 50 µl of transposition mix (100 nM TED1 (Illumina), 0.01% digitonin, and 0.1% Tween-20, in TD buffer

(Illumina)) and incubated at 37 °C for 30 min with 1000 RPM mixing. DNA was extracted from the reaction mixture with DNA Clean and Concentrator (Zymo Research, Irvine, CA, USA). DNA library was prepared using NEBNext® Ultra™ DNA Library Prep Kit for Illumina® (New England BioLabs) with five cycles of pre-amplification and three to seven cycles of PCR amplification. The amplified DNA library was purified with Zymo DNA Clean and Concentrator (Zymo Research), followed by two size-selection steps with SPRIselect (1:0.6 and 1:0.2 sample vol. to beads vol.; Beckman Coulter, CA, USA). The Omni-ATAC-seq libraries were sequenced using 50 bp single-end reads on the HiSeq 2500 (Illumina). The obtained sequence reads were mapped to the human hg19 genome by bowtie2. Reads mapped to the mitochondrial genome and duplicated reads were removed using removeChrom.py (Harvard ATAC-seq module) and samtools, respectively. Peak calling was performed using macs2 with a $10^{-5}$ cutoff $P$ value. The Omni-ATAC-seq was performed in two biological replicates.

**Quantitative reverse transcription PCR (qRT-PCR)**. Total RNA was reverse-transcribed using PrimseScript™ RT Master Mix (Takara Bio Inc.), followed by 10-fold dilution with EASY Dilution (Takara Bio Inc.). The real-time PCR were performed using TB Green® Premix Ex Taq™ II (Takara Bio Inc.) with the 7500 Fast real-time PCR system (Thermo Fisher Scientific Inc.). The thermal cycle condition was an initial step of 10 s at 95 °C, followed by 40 cycles of 3 s at 95 °C and 20 s at 62.5 °C. Gene expression changes were calculated using the $2^{-\Delta\Delta Ct}$ method. Primers used for the qRT-PCR are shown in Supplementary Table 1.

**Lentivirus preparation and transduction**. A GATA6 open reading frame was subcloned into the pCW57.1 vector using the Gateway LR reaction (Thermo Fisher Scientific Inc.). Doxycycline (Dox)-inducible GATA6 lentivirus vectors were produced by using the LV-MAX Lentiviral Production System (Thermo Fisher Scientific Inc.) according to the manufacturer's instructions.

**Cloning-based bisulfite sequencing**. The Dox-inducible GATA6 lentivirus vectors were transduced to HEK293T cells (BRC). After selection of the successfully transduced cells by culturing with 3 µg/ml puromycin, GATA6 were overexpressed by adding Dox at the final concentration of 500 ng/ml. After the 7 days of over-expression, the overexpression of GATA6 was shut off by removing the Dox. Genomic DNA was isolated using a NucleoSpin® Tissue Kit (Macherey-Nagel, Düren, Germany). Bisulfite C-T conversion were performed using EZ DNA methylation-Gold Kits (Zymo Research). The target genomic regions were amplified by 35 cycles of PCR using an EpiTaq™ HS (Takara Bio Inc.) according to the manufacturer's instructions. The PCR products were inserted into pTA2 plasmids using a TArget™ Clone Kit (Toyobo Co., Ltd., Osaka, Japan). The Sanger sequencing was done by Eurofins Genomics (Tokyo, Japan) with the M13-21 primer. The primers used to amplify the target region are shown in Supplementary Table 1.

**HaloTag pull-down assay**. HaloTag-fused- TET1 (FHC23878, Promega Corporation), -TET2(FHC22012, Promega Corporation), or -TET3(FHC30888, Promega Corporation) was co-transfected with pCDNA3.2-GATA6 to HEK293T cells (BRC) using Fugene HD (Promega Corporation). After 2 days from the transfection, the cells were collected and subjected to the HaloTag® Mammalian Pull-Down System (Promega Corporation) according to the man-ufacturer's instruction. The obtained sample were subjected to SDS-PAGE using NuPAGE Bis-Tris Gel (Thermo Fisher Scientific), followed by transfer to a PVDF membrane. The membrane was blocked by the Bullet Blocking One for Western Blotting (Nakarai Tesque Inc., Kyoto, Japan), incubated with an anti-GATA6 antibody in the Can Get Signal Solution 1 (Toyobo Co., Ltd.) for 12 h, and incubated with a secondary antibody in the Can Get Signal Solution 2 (Toyobo Co., Ltd.) for 1 h. The membrane was developed using ECL™ Prime (Cytiva, Marlborough, MA, USA), followed by image capturing by Fusin LS (Vilber Lourmat, Marne-la-Vallée, France). The antibodies used for the pull-down assay were shown in Supplementary Table 2.

**ChIPmentation**. ChIPmentation was performed using a ChIPmentation for Transcription Factor kit (Diagenode) according to the manufacturer's instructions. Briefly, the cells were fixed with 1% formaldehyde for 8 min. Chromatin was sheared by sonication using a Picoruptor® (Diagenode) for ten cycles and subjected to magnetic immunoprecipitation and tagmentation using an SX-8G IP-STAR® Compact Automated System (Diagenode). The sequencing libraries were amplified by nine cycles of PCR and cleaned up using AMPure XP beads (Beckman Coulter). The ChIPmentation libraries were sequenced using 150 bp paired-end reads on the HiSeq X (Illumina). The sequence reads were mapped to the human hg19 genome by bowtie2. Reads mapped to the mitochondrial genome and duplicated reads were removed using removeChrom.py and samtools, respectively. Peak calling was performed using macs2 with a $10^{-10}$ cutoff $P$ value. The antibody used for the ChIPmentation were shown in Supplementary Table 2. The ChIPmentation was performed in two biological replicates.

**Enzyme-linked immunosorbent assay (ELISA)**. Culture medium on the indi-cated day were collected. The medium was replaced with a fresh medium 3 days before the collection. According to the manufacturer's instruction, the medium was diluted 1000-fold and subjected to ELISA for AFP using the Human alpha Feto-protein ELISA Kit (Abcam, Cambridge, UK). Four biological replicates with two technical replicates were used.

**Tissue-specific gene enrichment analysis**. Upregulated genes between day 0 and day 28 were identified using the generalized linear model by *DBanalysis* function of the TCseq R package. For the upregulated genes, tissue-specific gene enrichment analysis was performed using enrichr with the ARCHS4 dataset.

**Functional analysis of differentially methylated regions**. GO analysis of dif-ferentially methylated regions was performed using *GREAT*[22]. Enriched GO lists were summarized based on Semantic Similarity by the GOsemSim R package.

**Screening of DNA methylation-regulating transcription factors**. TFBM over-representation analysis was performed using the *MotScr* function implemented in the InfiniumDiffMetMotR R package with the PWM database of Integrated ana-lysis of Motif Activity and Gene Expression changes of transcription factors (IMAGE)[45]. Out of the overrepresented motifs, the corresponding genes whose CAGE tag-per-million ≥50 at the time points where the TF binding motif was overrepresented were selected as DNA methylation-regulating transcription factors.

**Correlation matrix**. The correlation coefficient of all combinations of two clusters was computed using the M-values. The correlation coefficients were visualized as the correlation matrix heatmap. The clusters were ordered based on hierarchical clustering, which was calculated using the *hclust* and *dist* functions of the R stats package with the default settings.

**Functional analysis of differentially methylated regions**. Differentially methy-lated CpGs that were identified as ΔM > 2 and ±100 bp extended regions from the differentially methylated CpGs were used as differentially methylated regions. The differentially methylated regions were subjected to GREAT analysis using the *submitGreatJob* function implemented in the rGREAT R package with background data, which is with the regions extended ±100 bp for all methylation array probes. $Log_{10}$ FDR and the ratio between the numbers of hit regions and all differentially methylated regions of the Top10 overrepresented GOs (Biological Process) were visualized.

**Annotation of differentially methylated regions**. Gene promoters were defined as 1 kbp upstream and 200 bp downstream regions of genes in gencode human release version 19. The enhancers used in this study were FANTOM5 human phase 1 and 2 permissive enhancers. Non-promoter and non-enhancer regions were defined as unannotated regions. The complete overlap between uninherited demethylated CpGs and each regulatory region was counted.

**Coverage analysis of GATA6 ChIPmentation and Omni-ATAC-seq**. Bigwig Coverage files of CAGE ChIPmentation and Omni-ATAC-seq were computed using *bam2wig.py* with 1,000,000,000 wigsum (equals to coverage of 10 million 100 nt reads). The read coverage was visualized in the range between ±5 kbp from the demethylated CpGs using the *EnrichedHeatmap* function with the w0 mean model implemented in the EnrichedHeatmap R package.

**Permutation test between GATA6 ChIPmentation peaks and the demethy-lated regions of DE differentiation stage**. Differentially methylated CpGs that were identified as ΔM > 2 and ±200 bp extended regions from the differentially methylated CpGs were used as differentially methylated regions. Permutation tests were performed using *the permTest* function implemented in the regioneR R package with permutation 1000 times permutation.

**Statistics and reproducibility**. The numbers of biological and technical replicates are indicated for each experiment. The statistical tests used are shown in each analysis.

**Reporting summary**. Further information on research design is available in the Nature Research Reporting Summary linked to this article.

## Data availability

The datasets generated and analyzed during the current study are available in the NCBI Gene Expression Omnibus (GEO; http://www.ncbi.nlm.nih.gov/geo/) under accession number GSE163331. Source data underlying the main figures are presented in

Supplementary Data 1. The M-value data of GATA6-overexpressing HEK293T cells that support the findings of this study are available in Figshare (https://doi.org/10.6084/m9.figshare.18376592.v1).

## Code availability
Analysis codes and data are available in the Zenodo repository at the https://doi.org/10.5281/zenodo.6337878.

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

## Acknowledgements
We thank Chung-Chau Hon for useful advice in data analysis. We thank Jing-ru Li for the experiment support. We thank Hiroyuki Miyoshi and RIKEN BRC for providing lentivirus plasmids. We are grateful to RIKEN IMS, Laboratory for Comprehensive Genomic Analysis, for the Hiseq 2500 sequencing. This work was supported by Grant-in-Aid for Scientific Research (C) (19K08852) to T.S. from the Japan Society for the Promotion of Science. This work was also supported by a research grant from the Ministry of Education, Culture, Sport, Science, and Technology of Japan for the RIKEN Center for Integrative Medical Sciences.

## Author contributions
T.S. participated in the study's design, devised the methodology, performed the statistical analyses, carried out the molecular biology studies, acquired the funding, and drafted the manuscript. S.M., E.F., M.K., Y.M., Y.T., J.L., H.N., Y.N., and A.S. carried out the molecular biology experiments. H.S. helped to draft the manuscript, acquired the funding, and supervised the study. All authors read and approved the final manuscript.

## Competing interests
The authors declare no competing interests.

**Additional information**

