## [Peer Review File · Communications Biology]

Reviewers' comments:

Reviewer #1 (Remarks to the Author):

The authors present a study describing the DNA methylation changes that occur during the differentiation of induced pluripotent stem cells to hepatocyte-like cells. The work presented is of good quality, and the changes to the methylation status would be an interesting addition to our knowledge of hepatic development.

There are several drawbacks to the study. Firstly, the phenotypic analysis confirming the commitment to hepatocyte-like cells is not conclusive, making it hard to interpret the presented results. Secondly, the authors do not identify a transcription factor-specific mechanism of demethylation. Therefore, the results could be correlative with transcription factor-induced nucleosome repositioning, which has previously been described during liver development. Finally, the authors do not investigate the requirement for GATA6 to induce demethylation during endoderm formation. As such, while GATA6 binding is enriched at the demethylated regions, it may not be causative. Much of these concerns can be addressed by further data analysis of existing datasets, the specifics of which are included below:

Major points

1. It is surprising that the regions of demethylation post-endoderm formation do not feature FOXA2 and HNF4a binding sites at greater significance. This could be an interesting finding; however, it could also be that the cell population is not efficiently differentiating toward hepatocytes. The markers used to demonstrate hepatocyte differentiation follow the expected pattern, but are not conclusive markers of hepatic commitment, nor are they expressed at particularly high TPM.

Therefore, a greater number of hepatic enriched markers should be used to confirm an efficient differentiation process, e.g., by CAGE or qPCR analysis for ASGR1, APOB, ALBUMIN, HNF4A, and SLC10A1 mRNA expression at each timepoint. HNF4a is included in Supp. Figure 2B, but it is not possible to identify its expression pattern as there are many competing lines of similar color. Ideally, functional assays such as Albumin secretion should also be included.

2. The results show that GATA6 binds to loci that undergo DNA demethylation during endoderm formation. It is also true that GATA6 binds to thousands of sites that do not undergo DNA methylation and that the endodermal transcription factor network (EOMES, SOX17, FOXA2, etc.) has significant overlap in their binding profiles during this period. In the absence of mechanistic evaluation, by GATA6 knockout or knockdown, it is not possible to determine the role played by GATA6 in the demethylation process.

Using the datasets that they already have, the authors should give examples of GATA6 sites that are demethylated in the HEK overexpression experiment that are also bound by GATA6 and become demethylated during endoderm formation.

3. The temporal comparison of DNA methylation and chromatin accessibility profiles is interesting and could be used to establish mechanistic information about their relationship. For example, if the presented model is correct and demethylation follows chromatin accessibility, does demethylation result in more stable chromatin accessibility i.e., maintained rather than transient, compared to accessible regions that remain methylated. This will improve our mechanistic understanding regarding the consequence/function of the methylation changes.

Minor points

1. The authors have previously described interactions between RUNX1 and TET enzymes, indicating the capacity for RUNX1 to recruit enzymes related to the demethylation process. The authors do not present data or discuss whether GATA6 may function in a similar manner. Given the authors have these tools available, can they explain why this analysis was not included. Even if negative, this would provide novel mechanistic information regarding TF-induced demethylation

during liver development.

2. The authors state that the observed transient chromatin accessibility at a subset of loci that does not correlate with increased methylation is the result of additional TF binding interfering with ATAC-seq signal, rather than a real loss of chromatin accessibility (Line 380). This could also be the result of the slower dynamics associated with methylation than for chromatin remodeling. To substantiate this statement, can the authors provide examples of regions with reduced chromatin accessibility occurring at sites that maintain demethylated profile, GATA6 binding and CAGE read depth?

3. Line 261: Thus, our results suggest that GATA6 plays a major role in regulating DNA demethylation during DE differentiation.

As described above, the data is not sufficient to support this statement without mechanistic investigation.

4. Line 333: 'Because the methylation array analyses used in the present study do not distinguish between methylated cytosine and 5hmC, our results suggest that HNF4A-induced 5hmC is immediately converted to unmodified cytosine.'

This is one possible explanation that would require further investigation to support the stated conclusion.

5. Figure 3B – The authors state that GATA4 expression maximizes at 66 hours and GATA6 at 60 hours. From the presented datasets, GATA6 does indeed precede GATA4; however, the error bars at 66 hours make it hard to reliably conclude when GATA4 is maximally expressed – the text should be updated to reflect this. Also, given that the authors are comparing induction of the GATA factors, the same Y axis should be used for the graphs.

6. Figure 3C – The data should be presented using the same Y axis values.

7. Figure 4E/5B – It is unclear why the heatmaps have different scale bars next to them relating to each timepoint. The settings used to generate the heatmaps must be identical to allow for meaningful comparison. Can the authors clarify the meaning of these scale bars and correct the heatmaps if different settings have been used?

8. Figure 5C – The genomic loci of the viewer window and the scale for each dataset should be included. It is presumed the loci have been chosen as they follow the model outlined by the authors – can the authors additionally provide examples that neighbor hallmark endoderm or hepatic genes.

9. Figure 6A – The transcript data line appears identical between all subsets, is this correct?

Reviewer #2 (Remarks to the Author):

This is a very interesting and well-written manuscript describing the dynamics of DNA methylation changes during hepatocyte differentiation in vitro. The authors performed extensive analyses and identified several putative DNA methylation-regulating transcription factors, including GATA6. However, despite these extensive analyses, the manuscript in the present form is largely descriptive with a lack of mechanistic data.

Major concerns:

1. The mechanism of GATA6 induced DNA demethylation remains unexplored. All presented data only suggest the potential involvement of GATA6 and HNF4 in the DNA demethylation process.
2. In addition to the analysis of DNA methylation changes, it is necessary to investigate the state of DNA methylation and DNA demethylation machineries.

Minor comments:

1. The manuscript is too long, especially the "Discussion", with an excessive number of references.

Response to Comments

Referee #1:

The authors present a study describing the DNA methylation changes that occur during the differentiation of induced pluripotent stem cells to hepatocyte-like cells. The work presented is of good quality, and the changes to the methylation status would be an interesting addition to our knowledge of hepatic development.

There are several drawbacks to the study. Firstly, the phenotypic analysis confirming the commitment to hepatocyte-like cells is not conclusive, making it hard to interpret the presented results. Secondly, the authors do not identify a transcription factor-specific mechanism of demethylation. Therefore, the results could be correlative with transcription factor-induced nucleosome repositioning, which has previously been described during liver development. Finally, the authors do not investigate the requirement for GATA6 to induce demethylation during endoderm formation. As such, while GATA6 binding is enriched at the demethylated regions, it may not be causative. Much of these concerns can be addressed by further data analysis of existing datasets, the specifics of which are included below:

Response:

We thank the referee for recognizing the value of our study. We are also thankful to the referee for comments to extensively improve our manuscript. We have addressed all comments raised by Referee 1 in the response below and have modified the manuscript accordingly.

Major points

1. It is surprising that the regions of demethylation post-endoderm formation do not feature FOXA2 and HNF4a binding sites at greater significance. This could be an interesting finding; however, it could also be that the cell population is not efficiently differentiating toward hepatocytes. The markers used to demonstrate hepatocyte differentiation follow the expected pattern, but are not conclusive

markers of hepatic commitment, nor are they expressed at particularly high TPM.

Therefore, a greater number of hepatic enriched markers should be used to confirm an efficient differentiation process, e.g., by CAGE or qPCR analysis for ASGR1, APOB, ALBUMIN, HNF4A, and SLC10A1 mRNA expression at each timepoint. HNF4a is included in Supp. Figure 2B, but it is not possible to identify its expression pattern as there are many competing lines of similar color. Ideally, functional assays such as Albumin secretion should also be included.

Response:

In accordance with Referee #1's comment, we have added ASGR1, APOB, ALBUMIN, HNF4A, and SLC10A1 mRNA expression profiles during the hepatocyte differentiation in Fig. S1A.

Fig S1A

Line plot showing average TPM of each promoter of (A) pluripotent, DE, and hepatic differentiation marker genes; and (B) DNA methylation-related genes. The error bar is the standard deviation. X- and Y-axis are days of differentiation and average TPM, respectively. The experiment was done in triplicate.

We have also measured albumin level in the culture medium by ELISA. However, we did not detect albumin, which may be due to its low expression. Indeed, albumin mRNA expression was low in our CAGE analysis. To rule out experimental failure, we have also examined the AFP level. Medium AFP level increased between 14 and 28 days and decreased at 35 days, which is delayed from mRNA expression possibly due to the time required for translation and secretion. Thus, these results suggest that 28 days after our *in vitro* hepatocyte differentiation is the early stage of the mature hepatocyte. We have therefore added the result of ELISA for AFP in Fig. S1C and edited the main text as follows.

Page 9 Line 108

“In contrast, DE markers (*SOX17* and *FOXA2*) and hepatic markers (*HNF1B*, *HNF4A*, *ASGR1*, *APOB*, *SLC10A1*, *AFP*, *PAX6*, *PPARA*, *ALB*, and *CYP3A4*) were upregulated at day 7 and day 14 to day 28, respectively (Fig. S1A). The AFP protein level in the culture medium also increased between day 14 to day 28 and decreased at day 35, which was delayed from the change in mRNA level (Fig. S1B). We did not detect Albumin in the culture medium until day 35. Since *AFP* is known to be upregulated in immature hepatocytes and downregulated in mature hepatocytes, and *PAX6* and *ALB* are maturation markers of hepatocytes, our data confirmed that our *in vitro* differentiation mimics the *in vivo* hepatocyte development, until the early mature hepatocyte stage.”

Fig S1C

Albumin and AFP level in culture medium. X- and Y-axis days of differentiation and represent concentration, respectively. Each dot and error bar are mean and standard deviation. n.d. is “not detected” showing below the detection limit. The experiment was performed in four biological replicates with two technical replicates (* $p < 0.05$; ** $p < 0.01$; Student’s t-test).

2. The results show that GATA6 binds to loci that undergo DNA demethylation during endoderm formation. It is also true that GATA6 binds to thousands of sites that do not undergo DNA methylation and that the endodermal transcription factor network (EOMES, SOX17, FOXA2, etc.) has significant overlap in their binding profiles during this period. In the absence of mechanistic evaluation, by GATA6 knockout or knockdown, it is not possible to determine the role played by GATA6 in the demethylation process.

Using the datasets that they already have, the authors should give examples of GATA6 sites that are demethylated in the HEK overexpression experiment that are also bound by GATA6 and become demethylated during endoderm formation.

Response:

To corroborate our finding that GATA6 is involved in the DNA demethylation process, we performed DNA methylation analysis following GATA6 perturbation. To examine whether the demethylated regions induced by GATA6 overexpression can be recovered (re-methylated) by cutting off the GATA6 expression, we have transduced the GATA6 gene under a doxycycline-inducible promoter to HEK293T cells. To induce the

GATA6-induced DNA demethylation, we cultured the 293T cells with doxycycline for two weeks, then we further cultured the cells for two more weeks without doxycycline. Examining selected six regions where demethylation occurred in methylation array data using cloning-based bisulfite sequencing, we found the partial recovery of the demethylation. Thus, our new result corroborates the demethylation function of GATA6. We have added the result in Fig. 3D, Fig.S4, and the following text (Page 15, Line 201).

Page 15, Line 297

“Cloning-based bisulfite sequencing for ± 100 bp regions from four selected demethylated CpGs showed that turning the GATA6 overexpression off tended to partially recover the GATA6 overexpression-induced DNA demethylation (Fig. 3D; Fig. S4). These results indicate that GATA6 induces binding-site-directed DNA demethylation. In our previous report, RUNX1-mediated DNA demethylation was, at least partially, achieved by the association between RUNX1 and TET proteins.”

Fig.3D

Percentage of DNA methylation of ± 100 bp regions from four selected demethylated CpGs, analyzed by cloning-based bisulfite sequencing. Mock is mock control transduced and GATA6 is the doxycycline-inducible GATA6-transduced HEK293T cells. The cells were treated with doxycycline for two weeks (DOX+) and then cultured for 2 more weeks without doxycycline (DOX off). (* $p < 0.05$; ** $p < 0.01$, *** $p < 0.001$; Fisher's exact test).

Fig.S4

Fig. S4 Cloning-based bisulfite sequencing for GATA6 overexpression and withdrawing

DNA methylation patterns of ± 100 bp regions from four selected demethylated regions, analyzed by cloning-based bisulfite sequencing. Black and white circles indicate methylated and unmethylated cytosine of CpG, respectively. Horizontal lines represent the sequencing result of each sub-clone. Arrowheads indicate the CpGs that demethylated in methylation array analysis.

We have also analyzed overlap between GATA6 binding sites that are demethylated during DE differentiation and demethylated regions following GATA6 overexpression in HEK293T cells. We have identified 486 CpG sites that were demethylated during DE differentiation and overlapped with GATA6 ChIPmentation peaks. Of the 486 sites, 237 CpG sites (48.8%, P -value = 0.001, Permutation test) were also demethylated following GATA6 overexpression in HEK293T cells. We have added this result to the main text and as Fig S5E. We also provide an annotation list of the overlapped regions in Supplementary Table 1.

“The GATA6 ChIPmentation peaks at the 72-hour timepoint significantly overlapped with the regions demethylated during DE differentiation (486 regions: P -value = 0.001, Permutation test). Furthermore, of the overlapped regions, 48.8% (237 regions: P -value = 0.001, Permutation test) overlapped with the demethylated region by GATA6 overexpression (Fig. S5E). Thus, our results indicate correlation between GATA6 binding and DNA demethylation during DE differentiation.”

Fig S5E

Venn diagrams showing the overlap between GATA6 ChIPmentation peaks (GATA6) and demethylated regions of the DE differentiation stage (DE) (left) and the overlap between the intersect regions of the left Venn diagram ($DE \cap GATA6$) and demethylated regions induced by GATA6 overexpression in 293T cells (OE).

3. The temporal comparison of DNA methylation and chromatin accessibility profiles is interesting and could be used to establish mechanistic information about their relationship. For example, if the presented model is correct and demethylation follows chromatin accessibility, does demethylation result in more stable chromatin accessibility i.e., maintained rather than transient, compared to accessible regions that remain methylated. This will improve our mechanistic understanding regarding the consequence/function of the methylation changes.

Response:

To investigate whether demethylation leads to stable changes in chromatin state, we have analyzed the change in chromatin accessibility of ATAC-seq peaks that opened during 0 to 48 h. We found that the ATAC-seq peaks at demethylated regions stably kept higher chromatin accessibility, but ATAC-seq peaks that remain methylated

(hyper-methylation-maintaining regions) showed lower and more variable chromatin accessibility. We have therefore added the following text and Fig S6C.

Page 20, Line 278

“We also investigated the change in chromatin accessibility of ATAC-seq peaks, which opened in the period 0 to 48 h, dividing the peaks into demethylated and hyper-methylation-maintaining regions. Mean accessibility level was higher in peaks at the demethylated regions than those at the hyper-methylation-maintaining regions at all time points even before chromatin opening (iPS cells), and the deviation of chromatin accessibility tended to be higher in the hyper-methylation-maintaining regions, suggesting that hypo-methylation stabilizes the open chromatin status (Fig S6C).”

Fig S6C

(C) Violin plots showing chromatin accessibility of ATAC-seq peaks opened in the period 0 to 48 h. left and right panels show the accessibility of peaks overlapped with the demethylated regions and hyper-methylation-maintaining regions, respectively. X- and Y axis are DE differentiation timepoints and log Count Per Million (CPM). Dot, vertical bar, and gray shade at each data represent mean, S.D, and distribution of data.

Minor points

1. The authors have previously described interactions between RUNX1 and TET enzymes, indicating the capacity for RUNX1 to recruit enzymes related to the demethylation process. The authors do not present data or discuss whether GATA6 may function in a similar manner. Given the authors have

these tools available, can they explain why this analysis was not included. Even if negative, this would provide novel mechanistic information regarding TF-induced demethylation during liver development.

Response:

To examine the interaction between GATA6 and TET proteins, we co-overexpressed HaloTag-fused TET proteins and GATA6 in HEK293T cells followed by a HaloTag pull-down assay. The pull-down assay showed physical associations between GATA6 and TET proteins, suggesting the involvement of TET proteins in the GATA6 inducing binding site-directed DNA demethylation. We have therefore added Fig.3E and the following text.

Page 15, Line 201

“In our previous report, RUNX1-mediated DNA demethylation was, at least partially, achieved by the association between RUNX1 and TET proteins²⁵. Therefore, we examined the physical interaction between the GATA6 and TET proteins. Utilizing the pull-down assay between Halo-Tag-fused TET proteins and GATA6, we found an association between TET proteins and GATA6, suggesting that GATA6 recruits TET proteins to their binding sites (Fig. 3 E).”

Fig. 3E

2. The authors state that the observed transient chromatin accessibility at a subset of loci that does not correlate with increased methylation is the result of additional TF binding interfering with ATAC-seq signal, rather than a real loss of chromatin accessibility (Line 380). This could also be the result of the slower dynamics associated with methylation than for chromatin remodeling. To substantiate this statement, can the authors provide examples of regions with reduced chromatin accessibility occurring at sites that maintain demethylated profile, GATA6 binding and

CAGE read depth?

Response:

We thank the reviewer for pointing it out. We agree with the referee's comment. We have already mentioned the possibility of different reaction times between chromatin remodeling and DNA methylation in the Discussion of the original manuscript. (Page 36, Line 360 in the revised version).

Page 25, Line 355

“This sequential reaction may be merely due to differences in reaction times between chromatin remodeling and DNA demethylation.”

We have also shown genome browser screenshots of GATA6, SOX17, and FREM1 loci as examples of known GATA6 targets which are demethylated with chromatin opening during DE differentiation in Fig. 5C, Fig. S6B. The chromatin accessibility of these regions is prone to be maximum around 45 hours or 54 hours and reduced afterward.

Page 20, Line 272

“These demethylated regions coincident with chromatin opening during the DE differentiation stage included regulatory regions of known GATA6 targets, such as SOX17 and GATA6 (autoregulation) (Fig. 5C, Fig. S6B).”

Fig 5C

A screenshot of the genome browser showing DNA demethylated regions during DE differentiation stage, OmniATAC-read coverage, GATA6 ChIPmentation read coverage, and CAGE read coverage at GATA6 upstream region. The scale of each dataset is coverage of 10 million 100 nt reads. Red translucent rectangles represent demethylated regions. Enhancer track is based on GeneHancer database and enhancers overlapped with the demethylated region are shown as dark green. M-value profiles of each demethylated probe are shown above (x-axis: time point (hour), y-axis: M-value).

Fig S6B

Representative screenshots showing DNA demethylated regions showing in the genome viewer are GATA6 ChIPmentation read coverage and OmniATAC-read coverage at GATA6 upstream region. The scale of each dataset is coverage of 10 million 100 nt reads. Red translucent rectangles represent demethylated regions. Enhancer track is based on GeneHancer database and enhancers overlapped with the demethylated region are shown as dark green. M-value profiles of each demethylated probe are shown above (x-axis: time point (hour), y-axis: M-value).

3. Line 261: Thus, our results suggest that GATA6 plays a major role in regulating DNA demethylation during DE differentiation.

As described above, the data is not sufficient to support this statement without mechanistic investigation.

Response:

We agree with the reviewer's comment. We have therefore revised it as follows.

Page 18, Line 250

“Thus, our results indicate correlation between GATA6 binding and DNA demethylation during DE differentiation.”

4. Line 333: 'Because the methylation array analyses used in the present study do not distinguish between methylated cytosine and 5hmC, our results suggest that HNF4A-induced 5hmC is immediately converted to unmodified cytosine.

This is one possible explanation that would require further investigation to support the stated conclusion.

Response:

We have added the following description in the Discussion.

Page 23, Line 321

“Thus, these issues are unresolved at present and require further investigation.”

5. Figure 3B – The authors state that GATA4 expression maximizes at 66 hours and GATA6 at 60 hours. From the presented datasets, GATA6 does indeed precede GATA4; however, the error bars at 66 hours make it hard to reliably conclude when GATA4 is maximally expressed – the text should be updated to reflect this. Also, given that the authors are comparing induction of the GATA factors, the same Y axis should be used for the graphs.

Response:

In accordance with the referee’s comment, we have added the following text and have revised Figure 3A.

Page 14, Line 184

“*GATA6* and *GATA4* expression increased at 48 and 54 hours, respectively, after the induction of differentiation and continued to increase with differentiation (Fig. 3A).”

Fig. 3A

(A) qRT-PCR analysis for GATA4 (orange) and GATA6 (black). X- and Y-axes show time points of differentiation (hours from differentiation initiation) and fold-change (compared with 0 hours, log₂ scaled), respectively.

6. Figure 3C – The data should be presented using the same Y axis values.

Response:

We have adjusted the Y-axis ranges of Fig. 3C.

Fig3C

Distribution of enrichment score for the GATA6 binding motif within $\pm 5,000$ bp of methylated (left) and demethylated (right) CpG probes in GATA6-overexpressing 293T cells. X- and Y-axes show distance from probe CpG position and enrichment score, respectively. Horizontal lines are enrichment score = 0.

7. Figure 4E/5B – It is unclear why the heatmaps have different scale bars next to them relating to each timepoint. The settings used to generate the heatmaps must be identical to allow for meaningful comparison. Can the authors clarify the meaning of these scale bars and correct the heatmaps if different settings have been used?

Response:

Thank you for pointing out the unreadable expression. The value of those heatmaps is the mean coverage of 10 million 100nt reads. To clarify the meaning of the value, we have revised the method and the legends. We have also modified Fig. 4E and 5B, using the same dynamic ranges for each Figure, and added the legends for the color bars.

Page 35, Line 492

“Coverage analysis of GATA6 ChIPmentation and Omni-ATAC-seq

Bigwig Coverage files of CAGE ChIPmentation, and Omni-ATAC-seq were computed using *bam2wig.py* with 1,000,000,000 wigsum (equals to coverage of 10 million 100 nt reads). The read coverage was visualized in the range between ± 5 kbp from the demethylated CpGs using the *EnrichedHeatmap* function with the w0 mean model implemented in the EnrichedHeatmap R package.”

Fig. 4E

(E) Enrichment heatmap showing mean GATA6 ChIPmentation read coverage of 100 bp window at a range of ± 5 kbp from demethylated CpGs. Each time point is horizontally aligned and each of the UDRs are vertically aligned. Dark blue is low coverage and orange is high coverages.

Fig. 5B

(B) Heatmaps showing mean Omni-ATAC-seq read coverage of 100 bp window at a range of ± 5 kbp from demethylated CpGs. Each time point is horizontally aligned and each of the UDRs are vertically aligned. Red is higher coverage of Omni-ATAC-seq reads.

8. Figure 5C – The genomic loci of the viewer window and the scale for each dataset should be included. It is presumed the loci have been chosen as they follow the model outlined by the authors – can the authors additionally provide examples that neighbor hallmark endoderm or hepatic genes.

Response:

To make Figure 5C more meaningful, we have replaced the genome browser screenshot with the GATA6 locus, which includes the information of genomic loci and the scale. In addition, SOX17 and FREM1 loci were also shown in Fig. S6B. These revised figures are shown in our response to Minor Point 2 of Referee #1.

9. Figure 6A – The transcript data line appears identical between all subsets, is this correct?

Response:

We are grateful for the referee's comment. We have confirmed all analysis processes and have noticed a bug in the analysis code. Therefore, we have fixed the analysis code and have performed the analysis again. We have therefore revised the following text and Fig. 6A.

Page 21, Line 288

“Overall, the kinetics of GATA6 binding, chromatin accessibility, and transcription observed the same trends, regardless of the UDRs. While transcription levels and $-\Delta M$ value at the UDRs tended to increase after 48 h in accordance with GATA6 expression, GATA6 binding dramatically increased between 48 to 54 h and plateaued at 54 h, with a transient decrease at 66 h. Of note, chromatin accessibility increased in the period 0 to 48 h or 54 h and then decreased after the peaking, although DNA continued to be demethylated.”

Fig6 A

Line plots showing changes in each demethylated region's omics data. X-axis is log₂ fold-change (FC) for read coverages of ATAC-seq and ChIPmentation for GATA6 (left scale), and $-\Delta M$ -value (left scale); TPM for GATA6 expression (right scale). Y-axis represents the time points of the differentiation. CAGE based GATA6 expression profiles are identical among the panels.

Referee #2:

This is a very interesting and well-written manuscript describing the dynamics of DNA methylation changes during hepatocyte differentiation in vitro. The authors performed extensive analyses and identified several putative DNA methylation-regulating transcription factors, including GATA6. However, despite these extensive analyses, the manuscript in the present form is largely descriptive with a lack of mechanistic data.

We thank the referee for the positive assessment of our data. We are also thankful to the referee for the comments. We have addressed all comments raised by Referee 1 in the response below and have modified the manuscript accordingly.

Major concerns:

1. The mechanism of GATA6 induced DNA demethylation remains unexplored. All presented data only suggest the potential involvement of GATA6 and HNF4 in the DNA demethylation process.

Response:

In accordance with the comment, we performed additional experiments to verify our results. We believe these results corroborate our findings.

First, to investigate the possible molecular mechanism underlying DNA demethylation induced by GATA6, we co-overexpressed HaloTag-fused TET proteins with GATA6 in HEK293T cells followed by HaloTag pull-down assay. The pull-down assay showed physical associations between GATA6 and TET proteins, suggesting the involvement of TET proteins in the GATA6-induced binding site-directed DNA demethylation. We have therefore added Fig.3E and following text.

Page 15, Line 201

“In our previous report, RUNX1-mediated DNA demethylation was, at least partially, achieved by the association between RUNX1 and TET proteins. Therefore, we examined the physical interaction between the GATA6 and TET proteins. Utilizing the pull-down assay between Halo-Tag-fused TET proteins and GATA6, we found an association between TET proteins and GATA6, suggesting that GATA6 recruits TET proteins to their binding sites (Fig. 3 E).”

Fig. 3E

We have also performed DNA methylation analysis following GATA6 perturbation to corroborate our finding that GATA6 is directly involved in the DNA demethylation process. To examine whether the demethylated regions induced by GATA6 overexpression can be recovered (re-methylated) by cutting off the GATA6 expression. We have transduced GATA6 gene under a doxycycline-inducible promoter to HEK293T cells. To induce the GATA6-induced DNA demethylation, we cultured the 293T cells with doxycycline for two weeks, then we further cultured the cells for two more weeks without doxycycline. Examining four selected regions where demethylation occurred in methylation array data using cloning-based bisulfite sequencing, we found the partial recovery of the demethylation. Thus, our new result corroborates the demethylation function of GATA6. We have added the result in Fig. 3D, Fig.S4, and the following text (Page 15, Line 201).

Page 15, Line 197

“Cloning-based bisulfite sequencing for ± 100 bp regions from four selected demethylated CpGs showed that turning the GATA6 overexpression off tended to partially recover the GATA6 overexpression-induced DNA demethylation (Fig. 3D; Fig. S4). These results indicate that GATA6 induces binding-site-directed DNA demethylation. In our previous report, RUNX1-mediated DNA demethylation was, at least partially, achieved by the association between RUNX1 and TET proteins.”

Fig.3D

Percentage of DNA methylation of ± 100 bp regions from four selected demethylated CpGs, analyzed by cloning-based bisulfite sequencing. Mock is mock control transduced and GATA6 is the doxycycline-inducible GATA6-transduced HEK293T cells. The cells were treated with doxycycline for two weeks (DOX+) and then cultured for 2 more weeks without doxycycline (DOX off). (* $p < 0.05$; ** $p < 0.01$, *** $p < 0.001$; Fisher's exact test).

Fig.S4

Fig. S4 Cloning-based bisulfite sequencing for GATA6 overexpression and withdrawing

DNA methylation patterns of ± 100 bp regions from four selected demethylated regions, analyzed by cloning-based bisulfite sequencing. Black and white circles indicate methylated and unmethylated cytosine of CpG, respectively. Horizontal lines represent the sequencing result of each sub-clone. Arrowheads indicate the CpGs that demethylated in methylation array analysis.

2. In addition to the analysis of DNA methylation changes, it is necessary to investigate the state of DNA methylation and DNA demethylation machineries.

Response:

We have added DNA methyltransferases and TET gene expression profiles in Fig S1B. Also, we have added these results in the main text.

Page 10, Line 127

“Expression of DNA methyltransferases tended to decrease with differentiation, in line with the decline in the number of methylated probes (Fig. S1C). While the expression of TET1 decreased with differentiation, that of TET2 spiked on Day 14, when the number of identified demethylated probes was the highest. Thus, these data suggest that the bias toward demethylation depends on the balance of methylation and demethylation enzymatic activities.”

Fig S1C

Line plot showing average TPM of each promoter of pluripotent, DE, and hepatic differentiation marker genes (A) and DNA methylation-related genes (B). The error bar is the standard deviation. X- and Y-axis are

days of differentiation and average TPM, respectively. The experiment was done in triplicate.

Minor comments:

1. The manuscript is too long, especially the "Discussion", with an excessive number of references.

Response:

We have shortened the main text (mainly Discussion and Methods sections), making the total number of words 4,744. We have also reduced references to 37.

Reviewers' comments:

Reviewer #1 (Remarks to the Author):

The authors have responded to many of the concerns raised in the previous reviews. Unfortunately, a critical concern that was raised by all of the reviewers has not been sufficiently addressed in the revision. In order to interpret the results with any degree of confidence, requires that the differentiation protocol irrefutably and homogeneously generates cells with hepatocyte characteristics. If the cells do not meet this threshold then the role for GATA6 or DNA methylation may well be unconnected to any role during hepatocyte differentiation. The authors demonstrate that the cells they produce are unable to express characteristic markers of hepatocyte fate, such as albumin, at levels that have physiological relevance (supp Fig 1) and micrographs of the cells show that that morphological hepatocyte characteristics are absent (Fig 1). So, despite the elegant bioinformatic characterization of genome wide DNA methylation and ATAC-seq data sets, the conclusions drawn by the authors remain unsupported.

1. Line 56: Hepatocyte Nuclear Factors (HNFs) are not a family of proteins as described, but a group of unrelated transcription factors that were historically identified in the hepatocyte nuclei.

2. In supp Fig1A there appears to a dramatic loss of hepatocyte marker expression by day 28 which doesn't seem to fit with the authors description in the text (line 110). In fact the differentiation profile seems a bit unusual compared to published data from many other protocols including the timing of induction of these markers. The limitation in using a kit is that the exact protocol and constituents used during the differentiation are not defined; nevertheless, it would be helpful to have a better description of the methods used to differentiate the cells.

3. Line 116: It's concerning that Albumin couldn't be detected in the medium and that AFP appears to be at infinitesimal amounts (1ng/ml). It would suggest that the differentiations are very inefficient. Normal protocols would be expected to produce a minimum of 100 ng/ml of albumin by day 20 from a confluent monolayer of cells. However, since the figure legend doesn't match the figure, it is difficult to follow what the authors are referring to in the text. It also wasn't clear how the authors had measured the Albumin and AFP levels. All of this raises the question of how capable the differentiation protocol is of generating bona fide hepatocytes. Without a comparison to a standard, such as unplated primary human hepatocytes, it's not possible to interpret the results as presented.

4. Also, examining bulk RNA and protein levels is not sufficient on it's own because it doesn't inform how heterogeneous the differentiations are. It could just be that just a few cells express the hepatocyte markers at high levels or a lot of cells express the markers at low levels. For this reason, it is necessary to additionally include immunostaining for markers or use ssRNA-seq. Over 80% of cells should be HNF4A positive by day 20 of differentiation. I understand, that this seems like a lot to focus on, but if the efficiency of the differentiation protocol is not well described, it is not possible to interpret the data on methylation with any confidence because it could simply represent changes in DNA methylation status in unrelated cell populations that have adopted non-hepatocyte fates. It would also be helpful to measure the levels of non-hepatocyte markers, especially those that are in cells that express GATA4 and GATA6 such as cardiac myocytes and pancreatic cells (which are PAX6 positive).

5. Line 117: It is not clear why the authors used PAX6 as a marker for hepatocyte maturation, since this is not the case.

6. The cells shown in the phase contrast images in Figure 1 simply do not exhibit hepatocyte characteristics. They look liked a mixed epithelial population. Hepatocytes have a clear cuboidal appearance, prominent nucleoli, lipid droplets, and glycogen accumulation, all of which are absent from the cells shown in the image.

7. A whole body of literature that directly deals with the role of GATA6 during the differentiation of endoderm from iPSCs is missing and should not only be cited but discussed in the context of the

presented data. {Fisher et al., 2017, Biol Open, 6, 1084-1095; Tiyaboonchai et al., 2017, Stem Cell Reports; Shi et al., 2017, Cell Stem Cell, 20, 675-688.e6; Liao et al., 2018, J Clin Invest; Heslop et al., 2021, Cell Rep, 35, 109145}.

Reviewer #2 (Remarks to the Author):

Thanks for addressing previous comments. The new additions are helpful to the readers. The revised manuscript is very clear and provides valuable new information.

Response to Comments

Reviewer #1:

In order to interpret the results with any degree of confidence, requires that the differentiation protocol irrefutably and homogeneously generates cells with hepatocyte characteristics. If the cells do not meet this threshold then the role for GATA6 or DNA methylation may well be unconnected to any role during hepatocyte differentiation. The authors demonstrate that the cells they produce are unable to express characteristic markers of hepatocyte fate, such as albumin, at levels that have physiological relevance (supp Fig 1) and micrographs of the cells show that that morphological hepatocyte characteristics are absent (Fig 1). So, despite the elegant bioinformatic characterization of genome wide DNA methylation and ATAC-seq data sets, the conclusions drawn by the authors remain unsupported.

Response:

We are grateful to the reviewer for the valuable comments. First of all, we have corrected the overstatement of cell types. We have changed "definitive endoderm cells", "hepatoblasts", and "hepatocytes" to "definitive endoderm-like cells", "hepatoblasts-like cells", and "hepatocyte-like cells (HLCs)", respectively. Accordingly, we have revised the title to "Prediction of DNA Methylation-regulating Transcription Factors in an *in vitro* Hepatocyte Differentiation Model".

The hepatic differentiation protocol used in this study has been characterized in two earlier reports (Asplund *et al.*, *Stem Cell Reviews and Reports* 12 (1), 90-104 (2016); Ghosheh *et al.*, *Stem Cells International*, 2016, 8648356 (2016)). These reports showed that the hepatic differentiation protocol efficiently gives rise to near-homogenous hepatic cells, although it did not activate some mature hepatic functions such as albumin secretion. Therefore, to validate the characteristics of hepatic differentiation in our study, we have performed immunocytochemistry for HNF4 α . The immunocytochemistry has shown that over 96% of the cells were HNF4 α positive in day 21 and day 28 cultures. Although mRNA expressions of some of the mature hepatocyte marker genes were decreased between day 21 and day 28, other mature hepatocyte marker genes kept their high expression.

Furthermore, secret AFP was also detected in day 21 and day 28 culture media. Thus, the hepatic differentiation of this study essentially reproduced earlier reports and therefore suggests that day 28 HLCs were still fetal or immature hepatocytes as discussed

in the earlier report (Asplund *et al.*, *Stem Cell Reviews and Reports* 12 (1), 90-104 (2016)). We acknowledge the drawback of hepatic function-related genes expression of this differentiation protocol. Still, we believe that the hepatic differentiation protocol fulfills the aim of this study as we analyze the epigenetic change in hepatic differentiation.

Thus, we have extensively revised the Result part to precisely describe the characteristics of the hepatic differentiation of this study. Furthermore, we have discussed the limitation of the *in vitro* differentiation method and possibilities for improvement in the Discussion part.

We have addressed all comments raised by Reviewer 1 in the response below and have modified the manuscript accordingly.

1. Line 56: Hepatocyte Nuclear Factors (HNFs) are not a family of proteins as described, but a group of unrelated transcription factors that were historically identified in the hepatocyte nuclei.

Response:

We have edited the text to clarify that HNFs are not a family.

[Page 5, Line 55]

Several transcription factors (TFs), including c-Jun, members of the Hepatocyte Nuclear Factor (HNF), and GATA family genes, are known to play important roles in liver development and hepatocyte differentiation.

2. In supp Fig1A there appears to a dramatic loss of hepatocyte marker expression by day 28 which doesn't seem to fit with the authors description in the text (line 110). In fact the differentiation profile seems a bit unusual compared to published data from many other protocols including the timing of induction of these markers. The limitation in using a kit is that the exact protocol and constituents used during the differentiation are not defined; nevertheless, it would be helpful to have a better description of the methods used to differentiate the cells.

Response:

The hepatic differentiation protocol we used in this study has been evaluated in earlier reports (Asplund *et al.*, *Stem Cell Reviews and Reports* 12 (1), 90-104 (2016); Ghosheh *et al.*, *Stem Cells International*, 2016, 8648356 (2016)). We have cited these references

and have added a more detailed description of the protocol in the Results part.

[Page 8, Line 101]

We induced HLCs from human iPSCs *in vitro* using the Cellartis hepatocyte differentiation system (Takara bio), which is composed of three differentiation steps: iPSC to DE-like cell, DE-like cell to hepatoblast-like cell, and hepatoblast-like cell to HLC, followed by a maintenance culture (Fig. 1A and 1B). It has been reported that although this hepatic differentiation protocol does not activate several hepatic function-related genes, the efficiency of the hepatic specification is relatively high.

Furthermore, we have added the culture conditions in Fig. 1A to describe the differentiation protocol better.

[Fig. 1A]

(A) Schematic illustration of *in vitro* hepatocyte differentiation and culture condition of each step.

To precisely evaluate our hepatic differentiation characteristics, we have also analyzed TBX3 as a hepatoblast marker; AAT(SERPINA1), KRT18, CYP1A1 as hepatic maturation markers. Although a part of the hepatic maturation marker was decreased between day 21 and day 28, as reviewer #1 pointed out, other hepatic maturation markers have kept expressing. Based on these results, we extensively modified the "*DNA methylation dynamics throughout hepatocyte differentiation*" section of the Result part.

[Page 9, Line 111]

To evaluate the differentiation in detail, we analyzed the mRNA expression of differentiation markers. Concurrent with the decrease of the pluripotent markers, DE markers peaked at day 7, indicating the DE-like cell stage (Fig. S1C). On day 14, hepatoblast markers were upregulated, indicating the hepatoblast-like cell stage (Fig. S1C). Between day 14 and day 21, hepatic markers were upregulated, and a part of

them kept the high expression between Day 21 and day 28, although others were decreased (Fig. S1C). *Alpha-fetoprotein (AFP)*, produced by the fetal liver but not by the adult liver *in vivo*, increased to 4.6-fold greater expression from day 7 to 21 than the published fetal liver CAGE expression data²¹. Then *AFP* expression decreased to 49.2 % of fetal liver level from day 21 to day 28 (Fig S1C and S2A). AFP protein was also detected by immunocytochemistry of day 21 cells and Enzyme-Linked ImmunoSorbent Assay in the culture medium (Fig. 1E; Fig. S2B). Whereas, although expression of *Albumin (ALB)*, which is expressed in mature hepatocytes *in vitro*, was increased from day 14, it was much lower compared with the published hepatocyte, fetal liver, and adult liver (Fig S1C and S2A) and secreted Albumin was not detected in the media (data not shown).

[Fig. S1C]

(C) Line plot showing average TPM of each promoter of pluripotent, DE, and hepatic differentiation marker genes. The error bar is the standard deviation. X- and Y-axis are days of differentiation and average TPM, respectively. The experiment was done in triplicate.

We have also discussed the limitation of the protocol and the possible cause of the loss of expression of several hepatic markers between day 21 and day 28 in the Discussion part.

[Page 22, Line 306]

Expressions of several hepatic function-related genes, such as Albumin, were low or not even in the day 28 HLCs (Fig S1B and S2A). Furthermore, several hepatic markers were decreased between day 21 and day 28, when the cells were cultured in the maintenance medium. This expression decline is likely the same phenomenon as the rapid loss of hepatic functionality when primary hepatocytes are cultured *ex vivo*. These drawbacks were already pointed out in the previous reports^{19,20}. Thus, because the liability must be due to the differentiation protocol, the functionality may be enhanced by recently reported improved protocol or 3D organoid culture^{32,33}. Nevertheless, because hepatic differentiation was high efficiency and near-homogeneous (Fig1C and 1D), the protocol used in this study seems to recapitulate the key molecular events of hepatic differentiation, which fulfill the purpose of this study.

3. Line 116: It's concerning that Albumin couldn't be detected in the medium and that AFP appears to be at infinitesimal amounts (1ng/ml). It would suggest that the differentiations are very inefficient. Normal protocols would be expected to produce a minimum of 100 ng/ml of albumin by day 20 from a confluent monolayer of cells. However, since the figure legend doesn't match the figure, it is difficult to follow what the authors are referring to in the text. It also wasn't clear how the authors had measured the Albumin and AFP levels. All of this raises the question of how capable the differentiation protocol is of generating bona fide hepatocytes. Without a comparison to a standard, such as un plated primary human hepatocytes, it's not possible to interpret the results as presented.

Response:

Thank you for pointing out the unclear results. Regarding Albumin expression, the earlier reports in which the same hepatic differentiation protocol was used ((Asplund *et al.*, *Stem Cell Reviews and Reports* 12 (1), 90-104 (2016), Figure 4D) have shown very low or no expression of Albumin, consistent with our result. Considering the AFP expression described below, this protocol likely gives rise to fetal or immature hepatocytes. We have added this observation in the Result part.

[Page 10 Line 122]

Whereas, although expression of Albumin (ALB), which is expressed in mature hepatocytes, was increased upon differentiation from day 14, it was much lower than that of the published hepatocyte, fetal liver, and adult liver (Fig. S1C and S2A) and secreted Albumin was not detected in the media (data not shown). Thus, these results suggest that day 21 and 28 cultures correspond to the fetal or immature hepatocyte stage, consistent with the earlier report.

With regard to the AFP ELSA, we have skipped calculating the dilution factor. Therefore, we have recalculated the result with the dilution factor and normalized by time as per 24 hours. To more clarify the procedure, we have also revised the Methods part.

[Fig. S2B]

(B) AFP levels in culture medium. The medium was replaced with fresh medium three days before the collection. The X-axis is the days of differentiation. Y-axis is the mean concentration per 24 hours. Each dot and error bar are mean and standard deviation. n.d. is "not detected" showing below the detection limit. The experiment was performed in four biological replicates with two technical replicates (* $p < 0.05$; ** $p < 0.01$; One-sided Student's t-test).

[Page 33, Line 472]

Enzyme-Linked ImmunoSorbent Assay (ELISA)

Culture medium on the indicated day were collected. The medium was replaced with a fresh medium three days before the collection. According to the manufacturer's instruction, the medium was diluted 1000-fold and subjected to ELISA for AFP using the Human alpha Fetoprotein ELISA Kit (abcam, Cambridge, UK). Four biological replicates with

two technical replicates were used.

We also performed immunocytochemistry for AFP using day 21 cells. Consistent with the mRNA expression and secreted AFP, we have detected AFP proteins. We have added the result to Fig. 1E.

[Fig. 1E]

(E) Immunocytochemistry for AFP. The left, middle, and right images are nuclei staining by DAPI, AFP staining, and the merged image of the nucleus and AFP stainings. The scale bar is 50 μm .

We also have compared *AFP* and *ALB* expression with published primary hepatocyte, adult liver, and fetal liver CAGE data (Forrest. A. et al. Nature 507, 462-470 (2014)). The *AFP* expressions on days 21 and 28 were 4.6-fold greater and approximately half than that of the fetal liver, respectively. On the other hand, *ALB* expressions of the *in vitro* differentiation samples were much lower than any of the primary samples.

[Page 9, Line 117]

Alpha-fetoprotein (AFP), produced by the fetal liver but not by the adult liver *in vivo*, increased to 4.6-fold greater expression from day 7 to 21 than the published fetal liver CAGE expression data [Forrest. A. et al. Nature 507, 462-470 (2014)]. Then *AFP* expression decreased to 49.2 % of fetal liver level from day 21 to day 28 (Fig S1B and S2A).

[Page 10 Line 122]

Whereas, although expression of *Albumin (ALB)*, which is expressed in mature hepatocytes *in vitro*, was increased from day 14, it was much lower compared with the published hepatocyte, fetal liver, and adult liver (Fig S1B and S2A) and secreted Albumin was not detected in the media (data not shown).

[Fig. S2A]

(A) *AFP* and *ALB* CAGE expression data of in vitro hepatic differentiation time-course and published primary hepatocyte, adult liver, and fetal liver (Forrest. A. et al. Nature 507, 462-470 (2014)). Replicate number is abbreviated to the rep, hep 1: hepatocyte donor 1, hep 2: hepatocyte donor 2, hep 3: hepatocyte donor 3.

4. Also, examining bulk RNA and protein levels is not sufficient on its own because it doesn't inform how heterogeneous the differentiations are. It could just be that just a few cells express the hepatocyte markers at high levels or a lot of cells express the markers at low levels. For this reason, it is necessary to additionally include immunostaining for markers or use ssRNA-seq. Over 80% of cells should be HNF4A positive by day 20 of differentiation. I understand, that this seems like a lot to focus on, but if the efficiency of the differentiation protocol is not well described, it is not possible to interpret the data on methylation with any confidence because it could simply represent changes in DNA methylation status in unrelated cell populations that have adopted non-hepatocyte fates. It would also be helpful to measure the levels of non-hepatocyte markers, especially those that are in cells that express GATA4 and GATA6 such as cardiac myocytes and pancreatic cells

(which are PAX6 positive).

Response:

We have performed immunocytochemistry for HNF4 α on days 21 and 28. Analyzing the HNF4 α positive cell rate, we have found that over 96% of the cells were HNF4 α positive. On the other hand, we have not detected NKX2.5 and NKX2.2 expressing cells as the marker of cardiac myocytes and pancreatic cells, respectively. These results have indicated the high efficiency and near-homogenous hepatic differentiation. We have added the HNF4A immunostaining result.

[Page 9, Line 106]

Indeed, over 96% of the cells in days 21 and 28 cultures were HNF4 α positive without pancreatic or cardiac cell marker expressing cells (Fig. 1C and 1D; Fig. S1A).

[Figs 1C and D]

(C) Immunocytochemistry for HNF4 α . The left, middle, and right images are nucleus staining by DAPI, HNF4 α staining, and merged images of the nucleus and HNF4 α stainings, respectively. The scale bar is 50 μ m. (D) Bar plot showing the average rate of HNF4 α stained nuclei. The error

bar is s.d. The data were collected from four different fields of a culture well in two biological replicates.

[Fig. S1A]

(A) Immunocytochemistry for NKX2.2 (upper) and NKX2.5 (lower) in day 21 cells. The left, middle, and right images are nucleus staining by DAPI, NKX2.2 or NKX2.5 staining, and merged images of the nucleus and protein stainings, respectively. The scale bar is 50 μm .

5. Line 117: It is not clear why the authors used PAX6 as a marker for hepatocyte maturation, since this is not the case.

Response:

Thank you for pointing out our misunderstanding. We have removed PAX6 and alternatively added AAT(SERPINA1), KRT18, CYP1A1 as the hepatic maturation markers to Fig S1B (please see above).

6. The cells shown in the phase contrast images in Figure 1 simply do not exhibit hepatocyte characteristics. They look like a mixed epithelial population. Hepatocytes have a clear cuboidal appearance, prominent nucleoli, lipid droplets, and glycogen accumulation, all of which are absent from the cells shown in the image.

Response:

For day 28 HLCs, we have taken a better micrograph to clearly show the morphological characteristics, carefully adjusting phase contrast and focus (z-axis) settings. We also have added a higher magnification micrograph at the left bottom. Although prominent nucleoli and lipid droplets were unclear, day 28 HLCs were polygonal cuboidal shape and

had a large cytoplasmic-to-nuclear ratio, typical hepatic morphologies.

[Fig 1B]

(B) Representative micrographs of cells at each time point. The scale bar is 100 μ m. At the bottom left of day 28, a higher magnification image of the area surrounded by the dotted-square shows.

7. A whole body of literature that directly deals with the role of GATA6 during the differentiation of endoderm from iPSCs is missing and should not only be cited but discussed in the context of the presented data. {Fisher et al., 2017, Biol Open, 6, 1084-1095; Tiyafoonchai et al., 2017, Stem Cell Reports; Shi et al., 2017, Cell Stem Cell, 20, 675-688.e6; Liao et al., 2018, J Clin Invest; Heslop et al., 2021, Cell Rep, 35, 109145}.

Response:

We have added the suggested references and have discussed them in the Discussion part as follows.

[Page 25 Line 355]

GATA6 is reported to be a pioneer factor that directly binds to non-permissive heterochromatin and primes the opening of chromatin and histone modifications by interacting with the chromatin remodeling complex [Heslop et al., 2021, Cell Rep, 35, 109145]. DNA demethylation by GATA6 may be a step toward pioneering. On the other hand, how the pioneer factors recognize their target regions is not clear yet. Our finding that GATA6 binding regions were already slightly accessible in iPSCs may explain the mechanism (Fig. 5B).

[Page 27 Line 377]

GATA6 plays pivotal roles in endoderm cell development and pancreas and lung formations [Fisher et al., 2017, Biol Open, 6, 1084-1095; Tiyafoonchai et al., 2017, Stem Cell Reports; Shi et al., 2017, Cell Stem Cell, 20, 675-688.e6; Liao et al., 2018, J Clin Invest]. Therefore, GATA6 haploinsufficiency causes several diseases, such as pancreatic agenesis. Because the epigenetic function of GATA6 was also confirmed in

HEK293T cells, which are non-endodermal, by the artificial ectopic expression system, GATA6 mediated DNA demethylation may be associated with the other biological system and pathology of these diseases.

REVIEWERS' COMMENTS:

Reviewer #1 (Remarks to the Author):

I believe the authors have addressed my major concerns. While my concerns about cell morphology remain and the lack of Albumin expression is concerning, the presence of HNF4A in the majority of cells is very reassuring.